# UFD-2 is an adaptor-assisted E3 ligase targeting unfolded proteins

Doris Hellerschmied[1,5], Max Roessler[2], Anita Lehner[3], Linn Gazda[1], Karel Stejskal[1], Richard Imre[1], Karl Mechtler[1], Alexander Dammermann [2] & Tim Clausen [1,4]

Muscle development requires the coordinated activities of specific protein folding and degradation factors. UFD-2, a U-box ubiquitin ligase, has been reported to play a central role in this orchestra regulating the myosin chaperone UNC-45. Here, we apply an integrative in vitro and in vivo approach to delineate the substrate-targeting mechanism of UFD-2 and elucidate its distinct mechanistic features as an E3/E4 enzyme. Using *Caenorhabditis elegans* as model system, we demonstrate that UFD-2 is not regulating the protein levels of UNC-45 in muscle cells, but rather shows the characteristic properties of a bona fide E3 ligase involved in protein quality control. Our data demonstrate that UFD-2 preferentially targets unfolded protein segments. Moreover, the UNC-45 chaperone can serve as an adaptor protein of UFD-2 to poly-ubiquitinate unfolded myosin, pointing to a possible role of the UFD-2/UNC-45 pair in maintaining proteostasis in muscle cells.

[1] Research Institute of Molecular Pathology (IMP) Vienna BioCenter (VBC) Campus-Vienna-Biocenter 1, 1030 Vienna, Austria. [2] Max F. Perutz Laboratories (MFPL), University of Vienna, Doktor-Bohr-Gasse 9, 1030 Vienna, Austria. [3] Vienna Biocenter Core Facilities, Doktor-Bohr-Gasse 3, 1030 Vienna, Austria. [4] Medical University of Vienna Vienna BioCenter (VBC) 1030 Vienna, Austria. [5] Present address: Department of Molecular, Cellular and Developmental Biology, Yale University, New Haven, CT 06511, USA. Doris Hellerschmied, Max Roessler, and Anita Lehner contributed equally to this work. Correspondence and requests for materials should be addressed to A.D. (email: alex.dammermann@univie.ac.at) or to T.C. (email: tim.clausen@imp.ac.at)

Forming a functional muscle requires the activity of dedicated assembly factors that organize structural and motor proteins into the highly ordered structure of the sarcomere, the basic contractile unit of the muscle. A major player in this process is the UCS (UNC-45/Cro1/She4p) protein UNC-45, a myosin chaperone critical for the assembly of thick filaments[1–3]. The importance of UNC-45 for muscle development was first demonstrated in the nematode *Caenorhabditis elegans (C. elegans)*, where a number of UNC-45 *temperature-sensitive (ts)* mutants have been identified[1,4–6]. When grown at the restrictive temperature, *ts* worms display a reduced number of thick filaments in their body wall muscles and exhibit an *uncoordinated (unc)* phenotype[1]. Further studies in *Drosophila* and zebrafish established that UNC-45 (Unc45b in vertebrates) is important for skeletal and cardiac muscle development[2,3,7]. Specifically, UNC-45 was shown to promote the folding of the motor domain of myosin molecules[8–10]. In this process, UNC-45 acts as a co-chaperone together with the general chaperones Hsp70 and Hsp90[8,11,12].

Despite its critical role for muscle biogenesis and function, the activity of UNC-45 needs to be carefully controlled, as elevated levels of the myosin chaperone are linked to severe muscle distortion[13,14]. To regulate UNC-45 function, higher eukaryotes developed a tailored degradation system relying on a highly specialized ubiquitin ligase, UFD-2[15,16]. UFD-2 was originally identified as part of the ubiquitin fusion degradation (UFD) network in *Saccharomyces cerevisiae*[17]. Although UFD-2 could not initiate the ubiquitination of target proteins itself, the ligase can extend short ubiquitin chains previously added by its partner E3 enzyme UFD4[18]. To describe this unique ubiquitination activity, the term "E4 ligase" was introduced[18], with UFD-2 serving as a model for similarly acting enzymes[19]. Further studies revealed the collaboration of the UFD-2 E4 ligase with other E3 enzymes, for example in the ERAD pathway[20,21], where UFD-2 is associated with the DOA10 E3 ligase and the CDC48 ATPase, ubiquitinating and remodeling damaged proteins of the endoplasmic reticulum. The activity of UFD-2 has also been studied in context of the regulated degradation of selected substrates such as Mps1, a cell-cycle kinase, and proteins involved in DNA damage

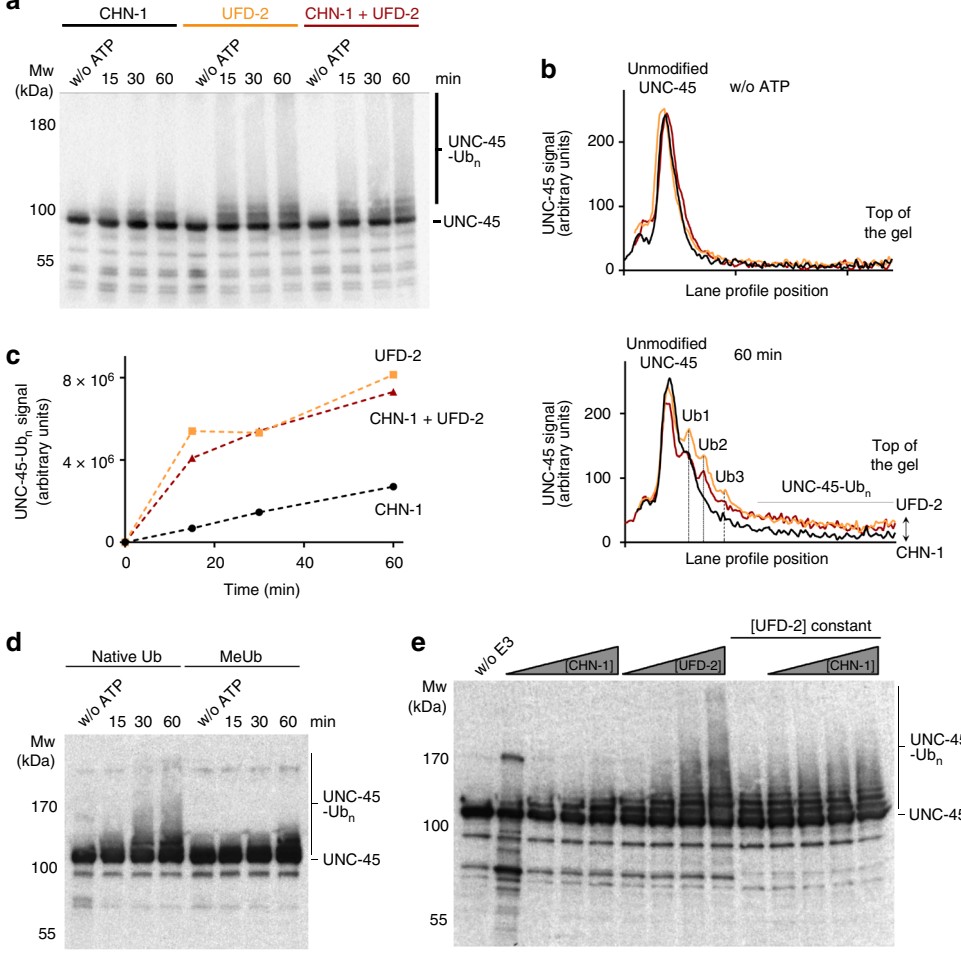

**Fig. 1** In vitro ubiquitination of UNC-45 by CHN-1 and UFD-2. **a** Time-course analysis of UNC-45 ubiquitination by CHN-1, UFD-2, or both E3 ligases. Reactions were incubated for 15, 30, and 60 min with ATP and analyzed by anti-UNC-45 western blot. **b** Quantification of the western blot lane profiles at time points 0 (without ATP) and 60 min. The two plots show the overlaid profiles upon addition of CHN-1 (black), UFD-2 (orange), and CHN-1/UFD-2 (red). **c** Quantification of the UNC-45($Ub_2–Ub_n$) signal. The corresponding ($Ub_2–Ub_n$) area of the reaction without ATP was used for background subtraction and is displayed as time point 0 for every reaction mix. **d** Ubiquitination of UNC-45 by UFD-2 using native or methylated ubiquitin (MeUb), with the latter preventing formation of poly-ubiquitin chains. Reactions were incubated for 15, 30, and 60 min with ATP and analyzed by anti-UNC-45 western blot. **e** Analysis of UNC-45 ubiquitination by using increasing concentrations (0.25, 0.5, 1, and 2 μM) of UFD-2 or CHN-1. To evaluate the possibility of a composite UFD-2/CHN-1 E4 activity, different amounts of CHN-1 (0.25, 0.5, 1, and 2 μM) were added to ubiquitination reactions containing 0.5 μM UFD-2. All assays were incubated for 60 min and were analyzed by anti-UNC-45 western blot

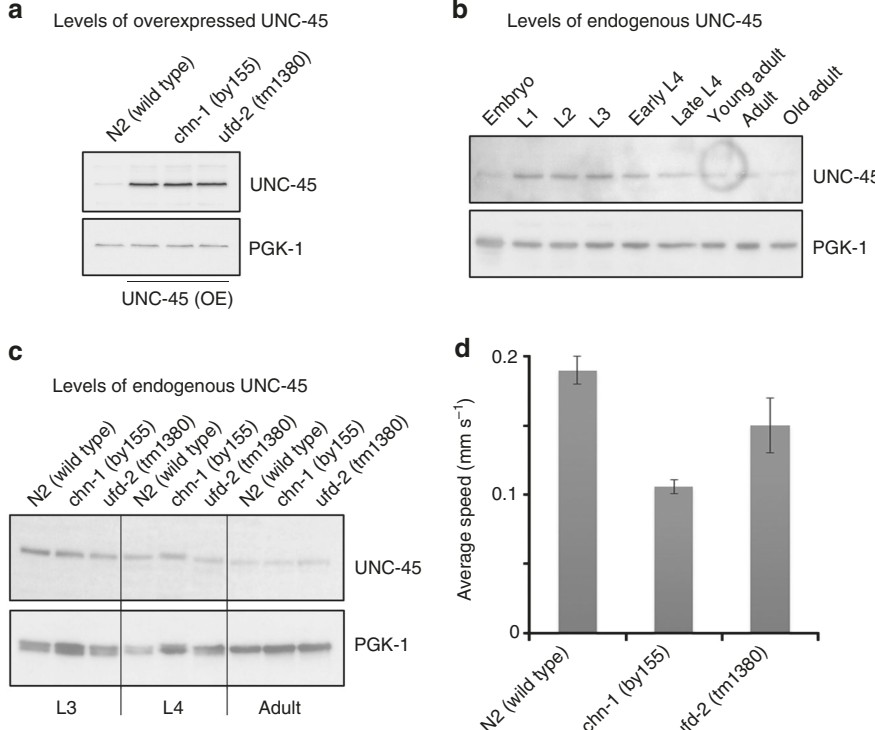

**Fig. 2** Functional interaction between UNC-45, CHN-1, and UFD-2 in vivo. **a** Comparison of UNC-45 protein levels when overexpressed in *unc-45(OE)*, *ufd-2(tm1380)*, and *chn-1(by155)* adult worms. PGK-1 was used as a loading control. **b** Western blot analysis showing the protein levels of endogenous UNC-45 at different stages of *C. elegans* development. **c** Western blot analysis showing protein levels of endogenous UNC-45 in wild-type, *ufd-2(tm1380)*, and *chn-1 (by155)* worms at the L3, L4, and adult developmental stages, respectively. **d** Average speed (mm s$^{-1}$) of indicated worms as determined in motility assays. Error bars represent sample standard deviation. Uncropped images of all western blots are shown in Supplementary Fig. 7

response[22–24]. In addition, UFD-2 can bind via its N-terminal domain to the UBL (ubiquitin-like) domain of Rad23 and Dsk2, substrate receptors of the 26S proteasome[25,26]. In conclusion, these studies suggest that UFD-2 acts downstream of E3 ligases extending the pre-assembled ubiquitin chains and preparing substrates for proteasomal degradation.

In contrast to the numerous E4 substrates of UFD-2, the myosin chaperone UNC-45 is the only protein described to be directly ubiquitinated by the ligase[15]. Together with CHN-1 (the *C. elegans* homolog of the mammalian CHIP protein) UFD-2 forms an E3/E4 ligase complex poly-ubiquitinating UNC-45. Importantly, the isolated UFD-2 and CHN-1 ligases were found to only attach single ubiquitin moieties to UNC-45; however, when applied together in vitro, UFD-2 develops its E4 activity promoting the polyubiquitination of the substrate[15]. Both enzymes, CHN-1 and UFD-2, contain a so-called U-box domain, a structural and functional homolog of the RING domain, that enables them to interact with ubiquitin-conjugating E2 enzymes and to transfer ubiquitin from the E2 to the substrate protein[27,28]. While CHIP proteins such as CHN-1 function as general quality-control factors marking misfolded, chaperone-bound proteins for proteasomal degradation[29], UFD-2 appears to be a more specific ubiquitination enzyme in higher eukaryotes important for UNC-45 regulation and muscle function. Consistent with such a specific role, the mammalian Ufd2a exhibits a remarkably distinctive expression pattern, showing the highest protein levels in skeletal muscle tissue[30]. Moreover, *ufd2a* knockout studies in mice demonstrated that the protein is essential for myocardial development[31].

Despite its important biological role and unique ubiquitination activity, the molecular mechanisms underlying UFD-2 ligase activity are largely unknown. To better understand how UFD-2 combines its general quality-control function in the ERAD and UFD pathways with its specific role in the proteolysis of the myosin chaperone UNC-45, we addressed its substrate-targeting mechanism. To this end, we analyzed the interplay of *C. elegans* UFD-2 with CHN-1 in modifying UNC-45. Our in vitro and in vivo data reveal that UFD-2 is a bona fide E3 ligase capable of poly-ubiquitinating unfolded protein segments and point to a role of the UFD-2/UNC-45 pair in myosin proteostasis.

## Results

**UFD-2 is an E3 ligase capable of poly-ubiquitinating UNC-45.** UFD-2 is considered to be the founding member of the E4 family of ubiquitin ligases that function as processivity factors elongating ubiquitin chains on certain substrate proteins[19]. The so far best-characterized E4 reaction is the polyubiquitination of UNC-45, mediated by the concerted activities of UFD-2 and CHN-1. To better understand the interplay of the E3/E4 ubiquitin ligases UFD-2 and CHN-1 in this process, we reconstituted the ubiquitination reaction in vitro using recombinantly expressed and purified components of the *C. elegans* system including the relevant E2 enzyme, UBC-2[15]. Upon incubating the substrate (UNC-45) with the isolated E3 ligases (UFD-2 or CHN-1) or the E4 complex (UFD-2/CHN-1 mixed in equimolar amounts), we followed the ubiquitination of UNC-45 by western blot analysis. Consistent with the previous results, CHN-1 alone could attach single ubiquitin moieties to UNC-45, but failed to generate poly-ubiquitin chains on the substrate (Fig. 1a). To our surprise, however, UFD-2 alone was able to poly-ubiquitinate the UNC-45 substrate, as clearly seen in the quantitative analysis of the E3 ligase assay (Fig. 1b, c, Supplementary Fig. 1). To test whether the attached ubiquitin is present in the form of linked chains or resulting from the multiple attachments of single subunits, we

carried out assays with a ubiquitin derivative preventing chain synthesis. As we could not detect any higher molecular-weight chains in this control reaction, we conclude that UFD-2 is a bona fide E3 ligase for the UNC-45 substrate (Fig. 1d). When we tested the influence of CHN-1 on this activity, we observed that adding the CHIP homolog did not alter the pattern or kinetics of the polyubiquitination reaction (Fig. 1a, e). Thus, the E3 activity of UFD-2 toward UNC-45 appears not to be influenced by CHN-1.

**Regulation of UNC-45 is not dependent on UFD-2 or CHN-1.** Given the discrepancy between these results and the reported activities of CHN-1 and UFD-2, we re-examined the effect of the two E3 ligases on UNC-45 levels in vivo. In order to reproduce the same experimental setup as in the previously published analysis[16], we first examined the role of UFD-2 and CHN-1 in animals overexpressing a FLAG-tagged version of UNC-45 in body wall muscles (strain *unc-45(OE)*). To determine the contribution of the two ubiquitin ligases in promoting the degradation of UNC-45, we analyzed worms lacking the functional CHN-

1 (a strain carrying the *chn-1(by155)* allele) or UFD-2 (a strain carrying the *ufd-2(tm1380)* allele) ligases. Western blot analysis of the worm lysates confirmed that UNC-45 levels were markedly increased in the *unc-45(OE)* strain (Fig. 2a). However, introducing the *unc-45(OE)* transgene into the *ufd-2(tm1380)* or *chn-1 (by155)* mutant background did not further increase the UNC-45 levels (Fig. 2a), suggesting that UNC-45 is not marked by these two ubiquitin ligases for proteasomal degradation.

To confirm these in vivo data, we also explored the effect of the two ubiquitin ligases on endogenous UNC-45. As it had been suggested that UFD-2 and CHN-1 mediate the developmental regulation of UNC-45, we monitored its protein levels at different larval and adult stages of the worm. Consistent with previous studies[16], western blot analysis revealed that endogenous UNC-45 is present in the highest amounts in the early larval stages (L1–L3) and is later downregulated to a lower steady-state level (Fig. 2b). Strikingly, however, the precisely timed regulation of UNC-45 was retained in both *chn-1(by155)* and *ufd-2(tm1380)* worms. As the endogenous UNC-45 protein levels were almost indistinguishable from wild type (Fig. 2c), neither UFD-2 nor

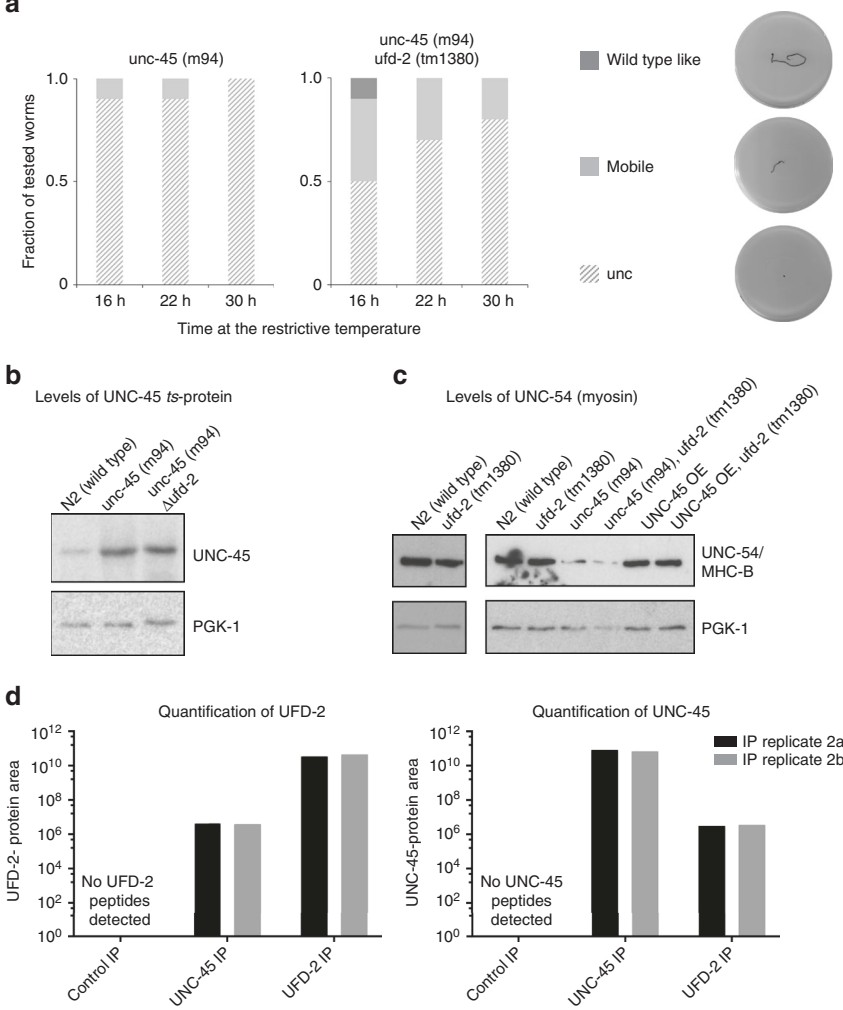

**Fig. 3** Functional interaction of UFD-2, UNC-45, and myosin in vivo. **a** Summary of crawling assays performed with *unc-45(m94)* and *unc-45(m94) ufd-2 (tm1380)* adult worms at the restrictive temperature of 23 °C for 60 min. Representative images show tracks generated by a single worm. For quantification, movement behavior was classified into three different categories - unc (uncoordinated phenotype), mobile (reduced motility), and wild-type-like motility. **b** Western blot analysis showing UNC-45 protein levels in wild-type, *unc-45(m94)*, and *unc-45(m94) ufd-2(tm1380)* adult worms. PGK-1 was used as a loading control. **c** Western blot analysis using an MHC-B specific antibody, showing levels of *C. elegans* muscle myosin in the indicated worm strains. **d** Quantification of the protein area of UNC-45 and UFD-2 in anti-UNC-45 and anti-UFD-2 IPs derived from parallel reaction monitoring (PRM). Two technical replicates are shown (two separate sets of IPs from the same lysate). No peptides were detected in control IPs

CHN-1 appears to be critical for the downregulation of UNC-45. To test whether the *ufd-2* and *chn-1* mutations exert an overall effect on muscle function, we performed a qualitative and quantitative analysis of the motility of young adult worms by tracking their movement in a defined arena. We found that both mutants exhibited clear motility defects as evidenced by a reduced average speed of individual worms compared to wild type

(Fig. 2d, Supplementary Fig. 2a). *chn-1(by155)* worms additionally displayed marked behavioral changes, including an increase in reversals and omega turns (two modes of changing direction in the arena) compared to wild-type animals (Supplementary Fig. 2b). These behavioral alterations point to defects beyond the locomotion system and may reflect a more global function of CHN-1, as expected for a CHIP homolog in a higher eukaryote[32].

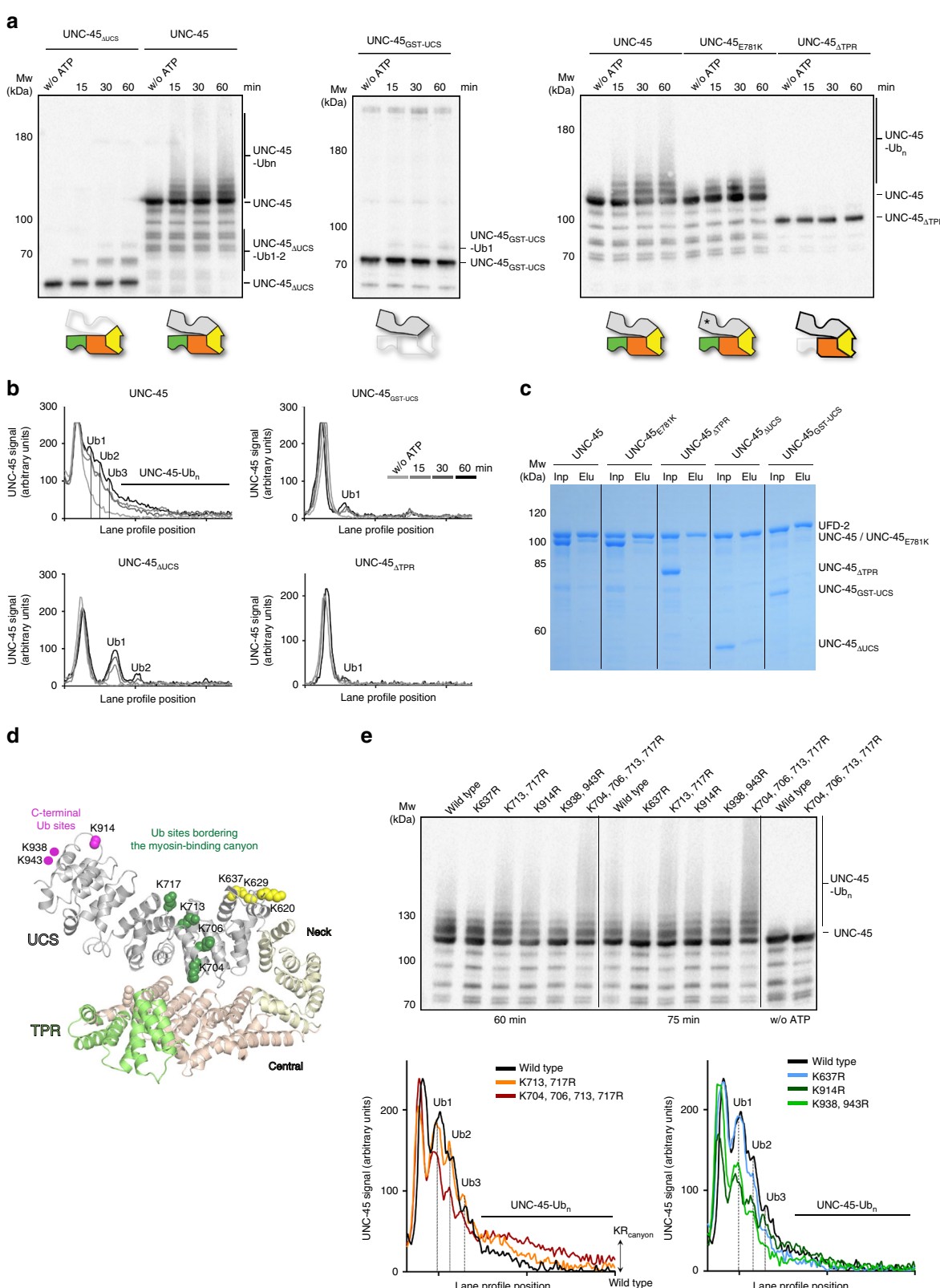

In conclusion, our in vivo data demonstrate that the CHN-1 and UFD-2 E3 ligases are not critical for the developmental regulation of UNC-45; however, they are important to develop and maintain full motility of *C. elegans* worms.

**Deletion of UFD-2 delays the UNC-45 *ts* phenotype**. As we could not observe a regulatory effect of UFD-2 on endogenous or overexpressed UNC-45 levels, we finally analyzed worms carrying the *ts* allele *unc-45(m94, mutation E781K)*, which originally revealed the functional connection between UFD-2 and UNC-45. According to Janiesch et al.[16], the absence of UFD-2 partially rescues the *unc* phenotype of the *ts*-mutant worms, most likely by stabilizing the corresponding UNC-45$_{E781K}$ mutant at the restrictive temperature. When analyzing the motility of *unc-45 (m94)* worms in crawling assays, we could confirm that the lack of UFD-2 partially rescues the temperature-dependent *unc* phenotype. Moreover, our detailed, time-resolved analysis revealed that the lack of UFD-2 delays the onset of the *unc* phenotype in *unc-45(m94)* worms, but cannot prevent it over time (Fig. 3a). When we quantified the UNC-45 levels in the analyzed strains at restrictive conditions (23 °C), we observed an increase of UNC-45$_{E781K}$ in the *unc-45(m94) ts*-mutant strain (Fig. 3b). A similar phenotype exhibiting elevated levels of the myosin chaperone was also observed for the related *ts*-mutant strains *unc-45(b131, mutation G427E)*, *unc-45(su2002, mutation L559S)*, and *unc-45 (e286, mutation L822F)*, pointing to a common compensatory mechanism (Supplementary Fig. 2c). Importantly, however, we did not detect any further stabilization of the UNC-45$_{E781K}$*ts*-mutant protein in the strain lacking UFD-2 (Fig. 3b). These in vivo data indicate that the recovery of motility upon loss of UFD-2 is not due to the stabilization of the UNC-45 protein.

We thus explored the effect of UFD-2 on the level of muscle myosin by a western blot analysis using an MHC-B-specific antibody. As previously described for the *unc-45(e286) ts*-mutant strain[13], we observed a strong decrease in myosin protein levels in *unc-45(m94)* worms. However, the levels of muscle myosin are not stabilized in the *ufd-2* deletion background (Fig. 3c). Although we could not detect any changes in the overall myosin protein levels, the level of functional myosin appears to be increased in *unc-45(m94) ufd-2(tm1380)* worms, as suggested by their partially restored motility (Fig. 3a). To address the potential role of UFD-2 in myosin quality control, we analyzed the interaction partners of UFD-2 under proteotoxic stress conditions. Using specific antibodies, we immunoprecipitated the protein from lysates prepared from heat-shocked worms. MS analysis of co-immunoprecipitated (coIP'ed) proteins revealed that UNC-45 and MHC-B (myosin heavy chain B, UNC-54) are interaction partners of UFD-2 in vivo (Supplementary Table 1). Moreover, when UNC-45 was immunoprecipitated from the same sample, UFD-2 as well as other muscle proteins could be detected in the elution fraction. Though we cannot exclude indirect binding, the interaction of UNC-45 and UFD-2, two non-sarcomeric proteins, could be functionally significant. To corroborate this interaction, we used the targeted MS approach of parallel reaction monitoring (PRM)[33] as a more sensitive method to quantify UNC-45 and UFD-2 peptides in two additional IP

samples (Fig. 3d). For this purpose, we selected peptides, identified in the original coIP experiment (Supplementary Table 4), for monitoring UNC-45 and UFD-2 abundance in the two samples. Our results strongly suggest that UFD-2 and UNC-45 directly interact in *C. elegans* muscle cells. Nevertheless, the small number of coIP'ed peptides compared to the bait protein indicate that the observed UFD-2/UNC-45 complex is relatively weak, pointing to a transient interaction of the E3 ligase and myosin chaperone.

**The UFD-2 E3 ligase targets the UCS domain of UNC-45**. To address the mechanism of how UNC-45 and UFD-2 collaborate with each other, we performed a detailed biochemical analysis of the UFD-2 ubiquitin ligase activity. Since UNC-45 is so far the only identified target that can be directly ubiquitinated by UFD-2 in vitro[15], we explored the specificity of this reaction by testing various potential substrate proteins. When the homologous human protein, HsUNC-45b, was incubated with *C. elegans* UFD-2 in our ubiquitination assay, only a faint band corresponding to mono-ubiquitinated HsUNC-45b could be detected (Supplementary Fig. 3a). Given the previous reports describing the role of UFD-2 in tagging unfolded protein substrates in the ERAD pathway[20,21], we used luciferase in the folded and unfolded state as additional model substrates. Notably, UFD-2 also failed to target both luciferase forms, as revealed by western blot analysis of the ubiquitination reactions (Supplementary Fig. 3b). These data show that despite the high enzyme concentrations used in the in vitro assay, *C. elegans* UFD-2 acts as an E3 ligase targeting the nematode UNC-45 protein in a specific manner.

We next used *C. elegans* UNC-45 as a model substrate to examine the molecular mechanisms of how UFD-2 recognizes specific peptide motifs and carries out the polyubiquitination reaction. To this end, we produced UNC-45 deletion constructs lacking the myosin-binding UCS domain (UNC-45$_{\Delta UCS}$) or the Hsp70/Hsp90-binding TPR domain (UNC-45$_{\Delta TPR}$). In addition, we generated a GST-fusion construct harboring the otherwise unstable UCS domain (UNC-45$_{GST-UCS}$). When we incubated UNC-45$_{\Delta UCS}$ with the UFD-2 E3 ligase, we could only observe residual ubiquitination activity, as reflected by the lack of any high-molecular weight-bands from poly-ubiquitinated proteins (Fig. 4a, b). Similarly, when we used the UNC-45$_{GST-UCS}$ or UNC-45$_{\Delta TPR}$ constructs as substrates, we could only observe a faint band resulting from mono-ubiquitinated proteins. To further characterize the interaction between UNC-45 and UFD-2, we performed a pull-down analysis using differently tagged UFD-2 and UNC-45 proteins (Fig. 4c, Supplementary Fig. 3c). These in vitro experiments demonstrated that full-length UNC-45 can directly interact with the UFD-2 ubiquitin ligase. Moreover, we noted that the UNC-45$_{GST-UCS}$ and UNC-45$_{\Delta TPR}$ deletion constructs no longer interacted with UFD-2, while deletion of the UCS domain resulted in a slightly weakened binding. Based on these data, the main binding site of UFD-2 seems to reside in the TPR domain of UNC-45, with the UCS domain stabilizing complex formation. In addition, the pull-down studies revealed that UFD-2 still interacts with the UNC-45$_{E781K}$*ts*-mutant, yet we observed a reduced ubiquitination of

**Fig. 4** Analysis of UFD-2 substrate specificity. **a** UFD-2-mediated ubiquitination of UNC-45$_{\Delta UCS}$, UNC-45$_{GST-UCS}$, UNC-45$_{\Delta TPR}$, and UNC-45$_{E781K}$ in comparison to the wild-type, full-length protein. **b** Plotted lane profiles of ubiquitination reactions shown in panel **a**. Mono- and di-ubiquitinated species are indicated for the deletion constructs. **c** Pull-down studies showing the interaction of UFD-2 with different UNC-45 proteins. **d** Cartoon representation of UNC-45 (PDB code: 4i2z) with TPR, central, neck, and UCS domains colored in green, orange, yellow, and gray, respectively. Ubiquitination sites identified in MS experiments are shown as colored spheres (green: residues bordering the myosin-binding canyon; magenta: residues near the C terminus of the UCS domain; and yellow: residues close to the flexible UCS loop). **e** Upper panel: 60 and 75 min time points of UNC-45 KR ubiquitination reactions with proteins that have been pre-incubated for 60 min at 30 °C were analyzed by anti-UNC-45 western blot. Lower panel: plotted lane profiles of the 60 min time point of the ubiquitination reactions, presented in two graphs for clarity

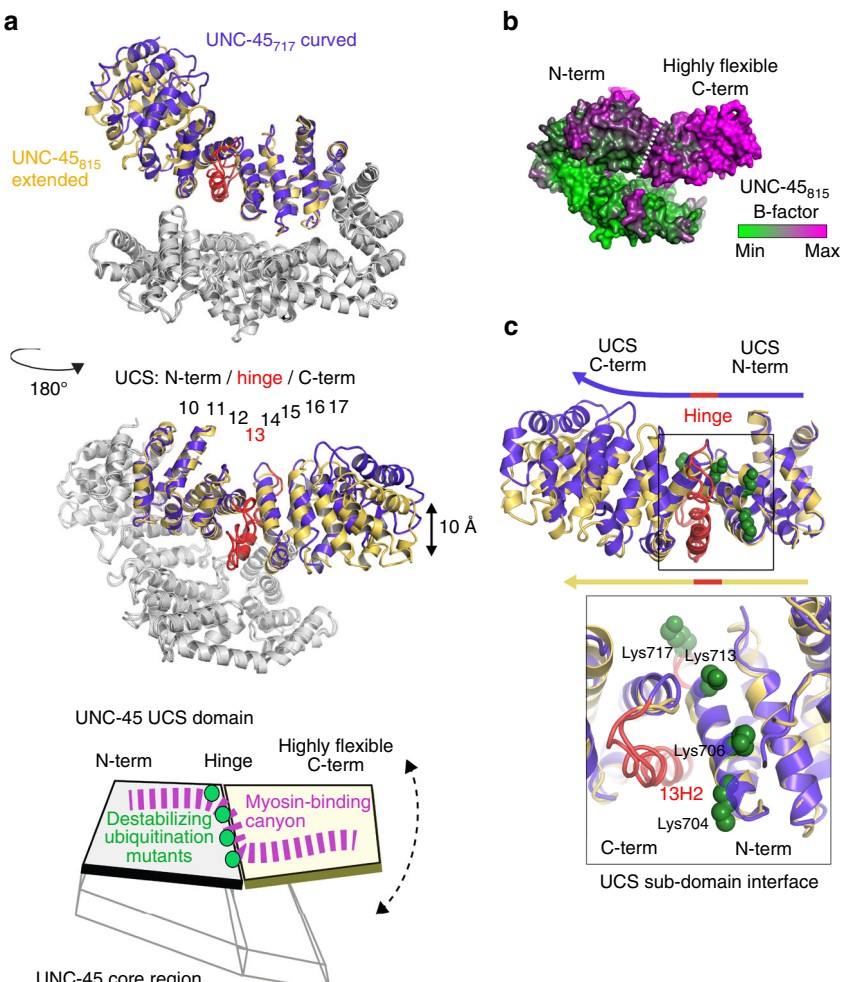

**Fig. 5** Location of ubiquitination sites on the flexible UCS domain of UNC-45. **a** Upper panel: structural alignment of UNC-45$_{815}$ (yellow) and UNC-45$_{717}$ (lilac) revealed that ARM repeat 13 (red) serves as a molecular hinge connecting the two UCS subdomains and allowing the domain to adopt distinct overall folds (straightened UCS superhelix in UNC-45$_{815}$, curved superhelix in UNC-45$_{717}$). The structure on top is depicted in the same orientation as illustrations in Fig. 4. Lower panel: schematic model of UNC-45. Green spheres indicate KR$_{canyon}$ mutations yielding a better UFD-2 substrate. **b** The subdomain architecture of the UCS domain is also seen when plotting the $B$ factors on the molecular surface (green, rigid; magenta, flexible). **c** Ribbon presentation of the UCS domain of UNC-45$_{815}$ (yellow) and UNC-45$_{717}$ (blue), shown in similar orientation as in Fig. 4. The zoomed-in window shows the KR$_{canyon}$ ubiquitination sites (green spheres) at the subdomain interface, as also indicated in **a**

this mutant protein, possibly pointing to changes in the targeted UCS domain (Fig. 4a). Taken together, the results of the interaction studies and ubiquitination assays indicate that the TPR domain of UNC-45 serves as the major docking site licensing UFD-2 to access and poly-ubiquitinate the adjacent UCS domain.

To identify the targeted lysine residues, we poly-ubiquitinated UNC-45 using UFD-2 and submitted the sample to mass spectrometric (MS) analysis. This analysis revealed a specific ubiquitination pattern with 8 from 10 identified sites residing in the UCS domain, the myosin-binding domain (Supplementary Table 2). As the 74 lysine residues of UNC-45 are distributed equally across the molecular surface (Supplementary Fig. 3d), the specific targeting of the UCS domain is a distinct feature of UFD-2. The ubiquitinated lysine residues are located in three regions of the UCS domain (Fig. 4d): one set of ubiquitination sites immediately borders the myosin-binding canyon in the center of the domain, whereas the remaining sites comprise lysine residues located in structurally flexible regions of the UCS domain at the very C terminus (K914, K938, and K943) and near the extended UCS loop (K620, K629, and K637), a structural motif also

implicated in myosin binding[11,34]. To further test whether the observed ubiquitination pattern reflects the targeting specificity of UFD-2 and is not a specific property of the substrate, we performed an MS analysis of UNC-45 ubiquitinated by the CHN-1 ubiquitin ligase. This analysis revealed that CHN-1 is highly promiscuous and ubiquitinates lysine residues spread over the entire surface of the UNC-45 molecule (Supplementary Fig. 3e). We thus conclude that UFD-2, in contrast to CHN-1, is a specific ubiquitin ligase directed against the UCS domain of UNC-45.

To further explore the substrate selectivity of UFD-2, we systematically exchanged the targeted lysine residues of UNC-45 with arginines (KR mutants) and analyzed the resultant mutants in ubiquitination assays. Unexpectedly, mutating lysine residues lining the myosin-binding canyon yielded a substrate that was better poly-ubiquitinated than wild-type UNC-45 (Fig. 4e and Supplementary Fig. 3f–h). Quantitative analysis of the ubiquitination profiles clearly revealed an increase in the high molecular weight poly-ubiquitinated UNC-45 for the K704/706/713/717 R (KR$_{canyon}$) mutant, while the K637R and K938/943 R mutants were not as strongly poly-ubiquitinated as wild-type UNC-45

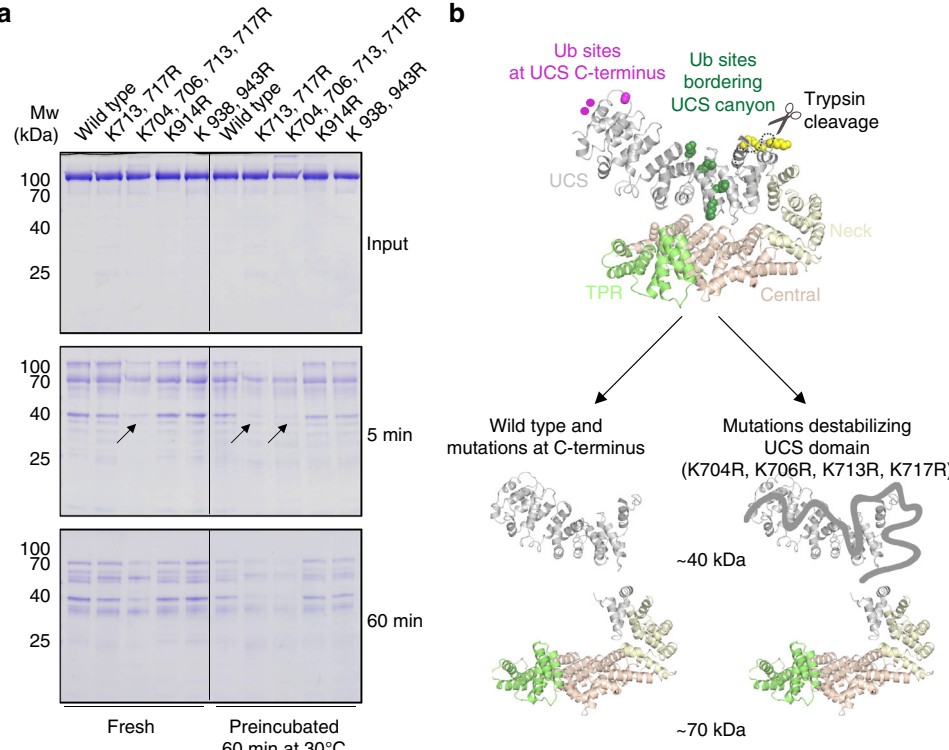

**Fig. 6** KR$_{canyon}$ mutations destabilize the UCS domain. **a** To visualize unfolded portions of the UNC-45 protein, limited proteolysis was carried out with trypsin. SDS-PAGE gels show trypsin digests of UNC-45 wild-type and mutant proteins pre-incubated for 60 min at 4 or 30 °C. Arrows highlight the disappearance of the 40 kDa fragment. **b** Degradation of the 40 kDa portion of UNC-45 illustrates the destabilizing effect of the UNC-45 KR$_{canyon}$ mutations on the C-terminal of the UCS segment

(Fig. 4e). These findings suggest that UFD-2 targets the UCS domain of UNC-45 in a topologically specific manner.

**Structural characterization of the targeted UCS domain**. To further characterize the structural motif targeted by the UFD-2 ligase, we performed a crystallographic analysis of the UCS domain of UNC-45. While extensive crystallization trials of the UCS domain containing various KR mutations were not successful, we succeeded in crystallizing the wild-type UNC-45 protein in a crystal form, capturing the UCS domain in a distinct state. The corresponding structure of UNC-45$_{717}$ (the number reflects the shortened crystallographic c-axis, when compared to the previously reported form, UNC-45$_{815}$[11]) was determined at 3.8 Å resolution (Supplementary Table 3, Supplementary Fig. 4) and exhibited a similar domain organization as UNC-45$_{815}$: the chaperone adopts a V-shaped structure with the myosin-binding UCS domain opposing the core region constituted by the TPR, central, and neck domains (Fig. 5a). Like UNC-45$_{815}$, the UNC-45$_{717}$ molecule formed protein filaments in the crystal lattice using the same 4-helix interface to connect molecules (Supplementary Fig. 4a, b). However, a pronounced structural change was seen in the UCS domain, a right-handed superhelix that was bent to different degrees reflecting two distinct conformational states (Fig. 5a, Supplementary Fig. 4c). Structural alignment of the two states revealed a bipartite organization of the UCS domain. Its N-terminal (ARM repeats 10–12) and C-terminal (ARM repeats 14–17) halves are connected by the irregular ARM repeat 13, the hinge of the domain. The disrupted interfaces between ARM repeats 12/13 and 13/14 appear to provide the conformational flexibility to allow for a 15° rotation of the C-terminal portion, a gross rearrangement resulting in a 10 Å shift of the C-terminal ARM repeat. The described subdomain architecture of

the UCS domain is also visible when plotting the thermal motion (B) factors on the molecular surface of UNC-45. Although the N-terminal UCS half is held in position by the underlying central domain, B-factor values increase abruptly from ARM repeat 14, the first segment of the C-terminal UCS subdomain (Fig. 5b, Supplementary Fig. 4d). Together, the crystallographic data underline the structural flexibility of the UCS domain, which can adopt discrete conformations. Strikingly, mapping the KR$_{canyon}$ mutations on the structure showed that the corresponding lysine residues cluster in a well-defined region of the UCS domain, contributing directly to the interface of the N-terminal and C-terminal subdomains (Fig. 5c). Accordingly, the introduced interface mutants that yielded better UFD-2 substrates may have a direct effect on the conformation and/or stability of the UCS domain.

**The UFD-2 E3 ligase acts on unfolded protein segments**. To biochemically address the stimulatory effect of the KR$_{canyon}$ mutations on UFD-2 ligase activity, we explored the extent to which these mutations influenced the organization and integrity of the UNC-45 substrate. For testing the stability of the different UNC-45 mutants, we performed limited proteolysis experiments. A trypsin digest of the wild-type protein led to the formation of two stable fragments of 70 and 40 kDa, indicative of a structured protein with two stably folded fragments (Supplementary Fig. 5a). Notably, the two fragments originate from a cut within the UCS loop (residues 602–630) after Lys620, as revealed by N-terminal sequencing. Consistent with this, an UNC-45 mutant lacking the UCS loop (UNC-45$_{\Delta UCSloop}$) is less efficiently cleaved by the protease (Supplementary Fig. 5a). We next incubated UNC-45 mutants with trypsin and monitored their degradation pattern over time by SDS-PAGE analysis (Fig. 6a, b). To this end, trypsin

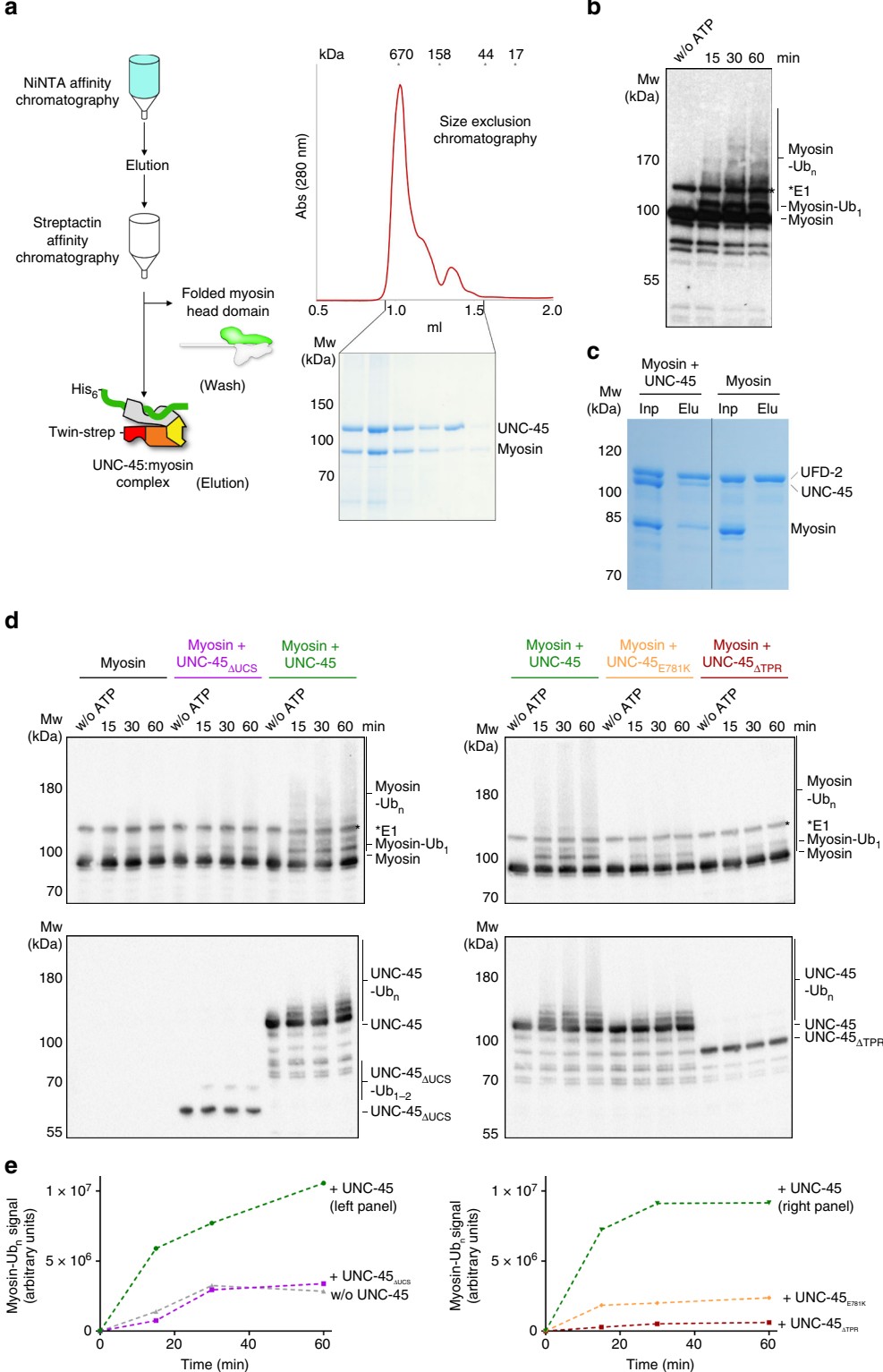

**Fig. 7** Ubiquitination of *C. elegans* myosin. **a** Purification of an UNC-45:myosin complex. The SEC profile and SDS-PAGE gel analysis of the final sample point to a stoichiometric UNC-45:myosin complex. **b** Time-course analysis of the ubiquitination of the purified UNC-45:myosin complex by UFD-2. **c** Pull-down analysis showing the interaction of UFD-2, UNC-45, and myosin, yielding a ternary complex. The control pull-down without UFD-2 is shown. **d** Ubiquitination of purified, heat-treated myosin (MHC-B) in the presence of UNC-45, UNC-45$_{\Delta UCS}$, UNC-45$_{E781K}$, and UNC-45$_{\Delta TPR}$. Reactions were incubated for 15, 30, and 60 min with ATP and a control reaction for 60 min without ATP, and analyzed by anti-His (upper panel) and anti-UNC-45 (lower panel) western blot. **e** Quantification of ubiquitinated myosin, using the same color code as in **d**. The ubiquitination signal observed without ATP was used for background subtraction and is displayed as time point 0 for every reaction mix

digests of freshly purified UNC-45 variants and proteins pre-incubated at 30 °C, the temperature applied in the ubiquitination assays were carried out in parallel. Strikingly, the C-terminal 40 kDa fragment of the $KR_{canyon}$ mutants was rapidly further degraded, suggesting that the $KR_{canyon}$ mutations destabilize the UCS domain, making its C-terminal part more susceptible to protease digestion.

To corroborate the limited proteolysis data, we performed circular dichroism (CD) measurements. The recorded melting curve of wild-type UNC-45 revealed an unfolding step at about 35 °C and a further transition occurring at 60–70 °C due to aggregation of the protein (Supplementary Fig. 5b–d). Strikingly, an analysis of the individual $T_m$ values (melting temperature of unfolding step) revealed a clear correlation between protein stability and the efficiency in being ubiquitinated by UFD-2. The most efficiently targeted UNC-45 $KR_{canyon}$ mutant proteins were also the most destabilized variants as indicated by a markedly lowered $T_m$ value (33 °C compared to 35 °C of wild type). In addition, we noted that deletion of the UCS domain generated a truncated protein having a higher $T_m$ value (38 °C) than the wild-type protein. These data indicate that the UCS domain represents the most unstable portion of the UNC-45 molecule. Altogether, the limited proteolysis experiments are consistent with a bipartite architecture of the UCS domain. Mutations of residues that are close to the molecular hinge connecting N-terminal and C-terminal half result in the most pronounced destabilization of the UCS domain. In particular, mutations of lysine residues located within the H3 helix of ARM repeat 12 or directly within the loop region preceding H2 of ARM repeat 13 reduced the stability of the domain (Fig. 5c).

In summary, structural and biochemical data demonstrate that the UCS domain of UNC-45 is a highly dynamic structure that is composed of a rigid N-terminal and a flexible C-terminal half. Mutating the interface of the two subdomains drastically destabilizes the UCS domain, most likely by detaching the C-terminal UCS portion from the remainder of the protein. As evidenced by the proteolysis and ubiquitination experiments, such loosening generates an unfolded protein stretch that can be targeted by UFD-2 with high efficiency.

**UFD-2 can utilize UNC-45 as adaptor to ubiquitinate myosin.** Based on the result that UFD-2 preferentially ubiquitinates unfolded parts of the UNC-45 myosin-binding domain, we asked whether unfolded myosin presented by the UCS domain of UNC-45 could be targeted as well. To prepare such a UNC-45:myosin complex, we co-expressed UNC-45 with the motor domain of the *C. elegans* muscle myosin MHC-B, a known substrate protein[1,35], in insect cells. A stoichiometric UNC-45:myosin complex could be isolated by performing a two-step affinity purification and subsequent size-exclusion chromatography (SEC) (Fig. 7a). The UNC-45:myosin complex with a predicted mass of 200 kDa eluted early, in stochiometric ratio, from the SEC column corresponding to an apparent molecular weight of about 670 kDa. To address the composition of this large complex, we analyzed the two components individually. While both proteins attained their compact, functional state at 4 °C, the heat-treated, presumably unfolded myosin eluted at a similar elution volume as the UNC-45:myosin complex (Supplementary Fig. 6a). Our SEC data thus suggest that myosin is captured in a largely unfolded form in the complex with UNC-45, thus reflecting the preference of the chaperone to bind to non-native protein segments as previously predicted from the shape of the myosin-binding canyon[11].

We next used the purified UNC-45:myosin complex as a substrate in the ubiquitination assay and could demonstrate that UFD-2 is able to poly-ubiquitinate myosin bound to UNC-45

(Fig. 7b, Supplementary Fig. 6b). To test whether the UFD-2-mediated ubiquitination is dependent on UNC-45, we compared the ubiquitination pattern of unfolded myosin in the presence and absence of its cognate chaperone. For this purpose, we reconstituted the UNC-45:myosin interaction in vitro by co-incubating MHC-B myosin with UNC-45 at higher temperatures to allow myosin unfolding and complex formation. We used a 1:1 mixture of UNC-45 and unfolded MHC-B directly as a substrate in the ubiquitination assays. These experiments showed that UFD-2 attaches single ubiquitin molecules to myosin even in the absence of UNC-45. Strikingly, however, in the presence of UNC-45, myosin was poly-ubiquitinated by UFD-2 (Fig. 7d, e). Consistent with these results, a direct interaction between UNC-45, myosin, and UFD-2 can be observed in pull-down studies (Fig. 7c, Supplementary Fig. 6c). Notably, the observed polyubiquitination was dependent on incubating UNC-45 and myosin at higher temperature (27 °C), as mixing at 4 °C prior to the ubiquitination reaction did not yield any poly-ubiquitinated myosin molecules (Supplementary Fig. 6d). These data provide compelling evidence that UNC-45 recruits UFD-2 to poly-ubiquitinate unfolded myosin.

When the deletion constructs UNC-45$_{\Delta TPR}$ and UNC-45$_{\Delta UCS}$, impaired in their interaction with UFD-2 and myosin, respectively, were mixed with MHC-B and tested in the ubiquitination assay, the formation of poly-ubiquitinated myosin was largely abolished. The same effect was observed for the UNC-45$_{E781K}$*ts*-mutant protein, suggesting that the mutant is not able to present its UCS domain (Fig. 4a) or myosin in a productive way to be targeted by UFD-2. To better characterize the effect of UNC-45 in promoting myosin ubiquitination by UFD-2, we performed a quantitative analysis of the ubiquitination assays (Fig. 7d, Supplementary Fig. 6eh). Although the full-length UNC-45 protein clearly supports attachment of the poly-ubiquitinated chains to myosin, the ubiquitination pattern of the deletion constructs are most similar to the reaction without the myosin chaperone. Together, these in vitro data demonstrate that the myosin-binding activity of UNC-45 as well as its ability to form a complex with UFD-2 are crucial to poly-ubiquitinate myosin. Moreover, the promiscuous CHN-1 E3 ligase could not be stimulated by UNC-45 to poly-ubiquitinate myosin (Supplementary Fig. 6f, g), highlighting the functional interplay of UNC-45 and UFD-2 in forming a composite ubiquitin ligase. In conclusion, our data reveal that UFD-2 is capable to poly-ubiquitinate unfolded myosin in a highly specific manner, employing the cognate myosin chaperone UNC-45 as an adaptor protein.

## Discussion

Muscle development and function relies on an intricate network of protein degradation and folding factors whose activity needs to be tightly regulated[36]. Focusing on the interplay between the myosin chaperone UNC-45 and the ubiquitin ligase UFD-2, we demonstrate that UFD-2 does not regulate the cellular levels of the chaperone. However, UFD-2 closely interacts with UNC-45, employing the chaperone as a co-factor to target unfolded myosin molecules. Collectively, these data reveal important mechanistic features of UFD-2, a central component of the protein quality-control system, and clarify its role as an adaptor-regulated E3 ligase.

The U-box containing UFD-2 is a specialized ubiquitin ligase involved in a plethora of degradation pathways in the cell. Notably, UFD-2 has been reported to function as an E4 elongation factor in modifying substrate proteins. As an E4 enzyme, UFD-2 acts downstream of certain E3 ligases such as UFD4 and DOA10, extending pre-assembled ubiquitin chains[18,20,37]. Clear

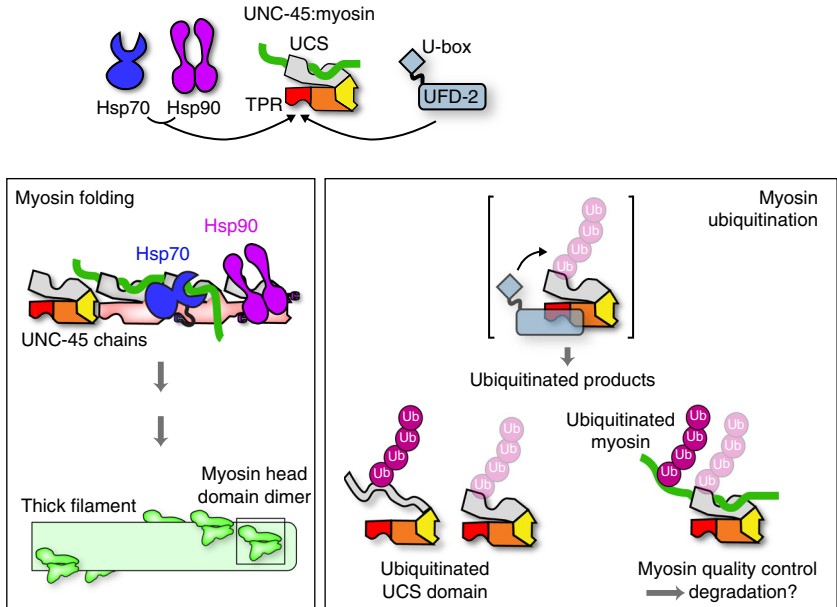

**Fig. 8** Collaboration of UFD-2 and UNC-45 with quality-control muscle myosin. UNC-45 is a myosin chaperone that teams up with Hsp70/Hsp90 to promote the folding and assembly of muscle myosin (left). Our data show that the UFD-2 ubiquitin ligase is another partner protein of UNC-45, redirecting the chaperone toward a ubiquitination pathway. Upon binding to its TPR domain (red), the UFD-2 E3 ligase gets properly positioned to poly-ubiquitinate a presented protein, either the UCS domain (gray) of UNC-45 itself or a UCS-bound myosin (green). Importantly, UFD-2 has a clear preference for marking unfolded proteins, as indicated

determinants of a UFD-2 substrate are so far elusive; however, its involvement in cellular quality-control pathways points to a broad range of substrate proteins. In contrast, studies addressing the immediate ubiquitin ligase activity of UFD-2 suggested a more specific function, linking UFD-2 to the degradation of the myosin chaperone UNC-45[15,16]. This ubiquitination reaction was reported to be carried out in conjunction with the CHIP homolog, CHN-1, that is critical to constitute the functional E4 ligase complex in vitro and in vivo[15,16]. Our data are in clear contrast to this model. We demonstrate that UFD-2 poly-ubiquitinates UNC-45 on its own, thus exhibiting bona fide E3 ligase activity in vitro. Modification of UNC-45 further does not depend on CHN-1, arguing against an E4 activity in this case. Moreover, global protein levels of UNC-45 were not affected upon deleting the ubiquitin ligases in C. elegans. As the regulatory proteolysis of a certain substrate protein is known to rely on E3 ligases that exhibit pronounced substrate specificity and operate in a precisely defined spatio-temporal manner as for example demonstrated for Aurora kinases[38], our data strongly suggest that UFD-2 and CHN-1 are not involved in the developmental regulation of UNC-45. In fact, the E3 ligase controlling the levels of UNC-45 remains to be identified to fully understand the biological role and regulation of this myosin-specific chaperone. While UNC-45 is not a major substrate for UFD-2 in vivo, we used UNC-45 as model to address the targeting mechanism of the UFD-2 U-box enzyme. Our data demonstrate that the protein site targeted and modified by UFD-2 needs to be partially unfolded. In vitro, the most preferred substrates comprised the unfolded UCS domain of UNC-45 itself and misfolded myosin bound to the UCS domain. We could further show that efficient ubiquitination of the UCS domain depends on the presence of the TPR domain, a well-known protein–protein interaction motif[39]. We thus propose that ubiquitination by UFD-2 proceeds in two steps. Upon binding to the TPR domain of UNC-45, the UFD-2 U-box domain, which exhibits pronounced en bloc flexibility[40], is then properly positioned to ubiquitinate the adjacent UCS domain and/or a UCS-bound ligand (Fig. 8).

To target a specific subset of substrate proteins, E3 ligases often make use of the so-called adaptor proteins. Prominent examples of such modular ubiquitination enzymes are the cullin-RING ligases and the CHIP ligase[29,41,42]. Our data show that the U-box containing E3 ligase UFD-2 is able to employ UNC-45 as a targeting factor to ubiquitinate the C. elegans muscle myosin, MHC-B. It thus appears that UFD-2 evolved a conceptually similar mechanism in utilizing molecular adaptors to select substrate proteins in a highly specific and regulated manner. In the case of the UNC-45 adaptor, substrate recognition and presentation is mediated by the UCS domain that harbors a specific binding site for myosin[8,11]. Our data suggest that myosin molecules, which are not present in their functional native state, are selectively targeted by the UFD-2/UNC-45 ubiquitination system. As the UNC-45 TPR domain is the major interaction site for both UFD-2 and Hsp70/Hsp90, it is likely that these interaction partners compete with each other, thereby determining the fate of the myosin bound to UNC-45. We propose that during development, the interplay with Hsp70 and Hsp90 is favored, promoting myosin folding and thick filament assembly. In contrast, during stress conditions, UNC-45 may preferentially interact with UFD-2 targeting, in this case, misfolded myosin molecules for degradation. Of note, E3 ligases, which are part of the cellular proteostasis network, frequently employ chaperones to select their substrates[43]. Ubr1, for example, requires Sse1 (S. cerevisiae Hsp70) to target misfolded proteins[44], while CHIP interacts with the general chaperone Hsp70 to forward aberrant client proteins for proteasomal degradation[29]. These data illustrate that pairs of chaperones and E3 ligases constitute central nodes in the quality-control network.

The ubiquitin proteasome system is critical for ensuring the proper assembly of muscle sarcomeres and maintaining their integrity, for example during stress situations or aging[36,45]. Among the many E3 enzymes, UFD-2 is highly abundant in muscle tissues. Its important role during development[31] was previously linked to the degradation of the myosin chaperone UNC-45[16]. Our in vitro and in vivo findings disagree with this

model. Our data rather suggest that UFD-2 functions as an E3 ligase that collaborates with the UNC-45 chaperone in removing unfolded myosin. Of note, the myosin quality-control system comprises several other members including MuRF E3 ligases[46], Atrogin-1[47], Ozz E3 ligase[48], and p97[49]. As we could not detect a stabilization of myosin in the *ufd-2* deletion strain, the distinct pathways ensuring myosin functionality appear to work in parallel compensating for each other. Such a redundant set of quality-control factors is a well-documented characteristic of cellular proteostasis, as seen for example in yeast, where the E3 ligases Ubr1 and San1 work in parallel, targeting unfolded proteins for degradation[44]. Similarly, substrates of the CHIP/Hsp70 system can be targeted by other E3 ligases when CHIP is absent[50]. Given this functional redundancy, it will be important to delineate further factors that operate in the same pathway as UNC-45 and UFD-2, keeping muscle proteins in shape.

## Methods

**Production of proteins from a bacterial expression system.** UNC-45$_{\Delta UCS}$ (1–521) and UNC-45$_{GST-UCS}$ (525–961) were cloned with a C-terminal STREP-tag or His$_6$-tag from cDNA and a synthesized UNC-45 gene (optimized for expression in *E. coli*, Supplementary Table 5) into pET21a and pGEX-6P-1, respectively. The different Lys to Arg (KR) point mutations as well as the E781K *ts*-mutation were introduced into full-length UNC-45 with the QuikChange Multi Site-Directed Mutagenesis Kit (Stratagene). The sequence of HsUNC-45b was synthesized and codon-optimized (Supplementary Table 5) for expression in *E. coli* (Life Technologies) and cloned into pET21a to generate a C-terminally His$_6$-tagged construct. Luciferase was cloned from pGEM®-luc (Promega) with a C-terminal STREP-tag into pET21a using accordingly designed DNA primers. CHN-1 and UFD-2 were cloned from *C. elegans* cDNA into the pET SUMO vector and pET28a to generate N-terminally His$_6$-SUMO-tagged and His$_6$-tagged construct, respectively. C-terminally Strep-tagged UFD-2, C-terminally Strep-tagged UNC-45$_{E781K}$, and C-terminally Strep-tagged and His$_6$-tagged UNC-45$_{\Delta TPR}$(135–961) were cloned into pCoofy. hUba1 in pET22b was a kind gift from Andrea Pichler. Primers are listed in Supplementary Table 6. Overexpression was carried out in *E. coli* BL21-DE3 cells (Invitrogen) for HsUNC-45b-His$_6$, luciferase-Strep, and His$_6$-SUMO-CHN-1 and in BL21-DE3-RIL (Stratagene) cells for His$_6$-UFD-2, UFD-2-Strep, His$_6$-SUMO-UFD-2, UNC-45$_{\Delta UCS}$-Strep/His$_6$ and UNC-45$_{GST-UCS}$-Strep/His$_6$, UNC-45$_{\Delta TPR}$-Strep/His$_6$, UNC-45-Strep/His$_6$ full-length, and UNC-45-Strep/His$_6$ point mutants. Protein expression was induced by adding IPTG to a final concentration of 250 μM, and incubating the cells for 5 h at 25 °C (HsUNC-45b-His$_6$, His$_6$-SUMO-CHN-1, and UNC-45$_{\Delta UCS}$-Strep/His$_6$), for 12 h at 25 °C (luciferase-Strep) or for 12 h at 18 °C (hUba1-His$_6$, His$_6$-UFD-2, UFD-2-Strep, His$_6$-SUMO-UFD-2, UNC-45$_{GST-UCS}$-Strep/His$_6$, and UNC-45$_{\Delta TPR}$-Strep/His$_6$). Expression of UNC-45 full-length and UNC-45 point mutants was induced with a final concentration of 100 μM IPTG for 12 h at 18 °C. The cells were harvested by centrifugation and lysed by sonication in 50 mM of Na$_2$HPO$_4$, pH 8.0, and 300 mM of NaCl (His$_6$-tagged and GST-tagged proteins) or 20 mM of Tris, pH 8.0, and 300 mM NaCl (Strep-tagged proteins). The tagged proteins were first purified by NiNTA, GST-Trap (GE Healthcare), or Streptactin (IBA) affinity chromatography, with the final elution buffer containing 150 mM imidazole, 10 mM glutathione, or 2.5 mM d-desthiobiotin, respectively. The His$_6$-SUMO-CHN-1 and His$_6$-SUMO-UFD-2 fusion proteins were incubated with His$_6$-SUMO-protease, and subsequently untagged CHN-1/UFD-2 was isolated in the flow-through fraction of an additional round of NiNTA affinity chromatography. All proteins were subjected to SEC using a Superdex 200 column (GE Healthcare) equilibrated with 20 mM Tris, pH 7.5 (pH 8.0 for UNC-45 full-length proteins), 150 mM NaCl.

**Production of proteins with the insect cell system.** The *C. elegans* MHC-B motor domain (1–790), referred to as myosin, was cloned from cDNA into pACEBac1 with a C-terminal fusion to a His$_6$-tag. UNC-45 was cloned into a pIDC derivative encoding an N-terminal Twin-Strep-tag (former One-Strep-tag) followed by a 3C protease cleavage site. The MultiBac was generated by Cre recombination (Geneva Biotech) and bacmids were generated by transposition of *E. coli* DH10EMBacY. Baculovirus was produced by transfection of SF9 cells (Expression Systems) with FuGENE HD (Promega). For expression, High Five cells (Expression Systems) at a density of $1 \times 10^6$ ml$^{-1}$ were infected with baculovirus (P1 amplification). The cells were incubated post infection for 96 h at 21 °C. The UNC-45:myosin complex was purified from infected cells by a two-step affinity purification procedure, first utilizing the His$_6$-tag on myosin and then the Twin-Strep-tag on UNC-45. The cells were collected by centrifugation, lysed by freeze–thaw, and resuspended in 50 mM Tris, pH 7.5, 300 mM NaCl (buffer A), containing 20 mM imidazole. The cleared cell lysate was loaded onto a NiNTA column (GE Healthcare) equilibrated with buffer A. After washing the column, myosin and associated UNC-45 were eluted with buffer A,Data collection and refinement statistics are containing 150 mM imidazole. The elution was

concentrated and loaded onto a Streptactin column (IBA), equilibrated with buffer A. Upon washing the column, the protein complex was eluted with 20 mM Tris, pH 8.0, 150 mM NaCl, and 2.5 mM d-desthiobiotin. Finally, the UNC-45:myosin complex was loaded onto a Superdex 200 column equilibrated with 20 mM Tris, pH 8.0, 150 mM NaCl. To obtain folded myosin, the wash fractions from the Streptactin purification step, containing myosin, were applied to a Superdex 200 column equilibrated with 20 mM Tris, pH 8.0, 150 mM NaCl. To obtain a SEC standard for an unfolded myosin, the folded protein (5 μM) was incubated for 60 min at 4 °C or 27 °C and applied to a Superdex 200 column in the same buffer.

**Crystallization and structure determination.** UNC-45 crystals were obtained by the sitting-drop vapor-diffusion method, upon mixing 2 μl of protein (200 μM) with 0.4 μl of additive (0.5 M NaF) and 0.5 μl reservoir solution (0.1 M HEPES pH 7.0, 10% PEG 8000, 12% ethylene glycol). Crystals grown at 19 °C were mounted in 90°-bent loops, shortly incubated in the crystallization solution containing 25% ethylene glycol, and flash-frozen in liquid nitrogen[11]. Compared to the initially characterized crystals (UNC-45$_{815}$), the new crystal form UNC-45$_{717}$ had a shortened *c*-axis, as reflected by the different unit-cell parameters of the P6$_1$22 space group (86.2, 86.2, 716.5 Å versus 86.6 86.6, 815 Å). Diffraction data were collected at the ESRF beamline ID 23-1 (wavelength: 0.979 Å), integrated with DENZO and scaled with SCALEPACK[51]. The structure of the UNC-45$_{717}$ crystal form was solved by molecular replacement, which was carried out with PHASER using the core region, i.e., TPR, central, and neck domains of UNC-45$_{815}$ (PDB code: 4i2z) as search model[52]. Refinement of the partial model with CNS yielded an improved electron density map that allowed incorporation of large parts of the UCS domain[53,54]. Data collection and refinement statistics are summarized in Supplementary Table 3. The final structure was refined with PHENIX at 3.8 Å resolution to an *R*-factor of 29.8% ($R_{free}$ value of 31.9%)[55]. Structure figures and distance measurements were generated with PYMOL[56]. Ramachandran statistics were computed with PHENIX (residues in favored regions: 89.9%; outliers: 2.2%).

**CD spectroscopy.** CD spectra were recorded at 0.7 mg ml$^{-1}$ protein concentration in 20 mM Tris pH 8.0, 150 mM NaCl using a Chirascan plus CD spectrometer (Applied Photophysics) connected to a temperature control unit (200–260 nm, 20–75 °C). Melting temperatures were determined by analyzing the data with the Global3 evaluation software.

**Limited proteolysis.** UNC-45 proteins were incubated with trypsin (Roche) at a 1/100 (w/w) ratio for the indicated times. Reactions were stopped by mixing with SDS-PAGE loading buffer and resolved on SDS-PAGE gels. N-terminal sequencing of selected protein bands was performed by Alphalyse.

**Ubiquitination assays and western blot analysis.** Ubiquitination assays contained 60 μM ubiquitin or methylated ubiquitin (BostonBiochem), 100 nM E1 (hUba1), 0.6 μM E2 (UBC-2), 1 μM E3 (UFD-2 or CHN-1), and 1 μM substrate protein (UNC-45, UNC-45 KR-mutants, UNC-45$_{E781K}$, UNC-45$_{\Delta UCS}$, UNC-45$_{GST-UCS}$, UNC-45$_{\Delta TPR}$ HsUNC-45b, and luciferase or MHC-B head domain) unless otherwise indicated in figure legends. For Fig. 4e, UNC-45 proteins were pre-incubated for 60 min at 30 °C to destabilize the UCS domain prior to the ubiquitination reaction. Unfolded luciferase and MHC-B/UNC-45 complexes were generated by incubation for 30 min at 50 °C and 60 min at 27 °C, respectively. All ubiquitination assays were performed in 20 mM Tris, pH 7.5, 50 mM NaCl, 1 mM DTT at 30 °C for the indicated times. Reactions were started by adding ATP/Mg (OAc)$_2$ to a final concentration of 10 mM and stopped by adding SDS-PAGE loading buffer. Reaction mixes were resolved on SDS-PAGE gels and stained with Coomassie or transferred to a PVDF membrane. Ubiquitination products were detected by performing western blot analysis using substrate-specific antibodies. anti-UNC-45b (peptide-specific antibody HPA017861, Sigma, used 1:1000), anti-Strep (monoclonal anti-Strep, Qiagen, used 1:1000), anti-Penta-His (Qiagen, used 1:1000), and anti-CeUNC-45 (specifically generated against the full-length protein, used 1:10,000). Secondary antibodies conjugated to horseradish peroxidase (Cell Signaling) were used in conjunction with Amersham ECL Western Blotting Detection Reagent (GE Healthcare). Protein bands were visualized using the ChemiDoc MP Imaging system (Bio-Rad), taking care to remain within the linear dynamic range of the detector.

**Quantification of in vitro ubiquitination products.** Quantification of in vitro ubiquitination reactions was performed on western blots developed using the ChemiDoc MP Imaging system (BioRad), which is designed for quantitative analysis of chemiluminescent signal. To additionally ensure that the quantified western blot signal is within the linear range of detection, we generated standard curves using a dilution series of the respective ubiquitination reaction, which was incubated for 180 min at 30 °C. The signal was quantified in ImageLab (Biorad) by drawing a box of the same size at the center of each lane showing ubiquitinated protein. The same size box was used to measure the background signal that was subtracted. The standard curve was generated in Prism (GraphPad) using the linear regression curve fitting model. The ubiquitination reactions in Supplementary Figs 1, 3 and 6 were quantified in the same manner. The thereby acquired HRP signals are within the linear range of the western blot as demonstrated by plotting

the obtained quantified signals onto the graph of the standard curve. To directly compare the distinct species of ubiquitinated protein ($-Ub_1$, $-Ub_2$, $-Ub_3$, $-Ub_n$), lane profiles of the western blots were generated for each reaction. A line was drawn at the center of the lane and the profile was plotted in Fiji[57]. Overlays of the lane profiles were generated in Prism (GraphPad). In case of an offset between different lanes, the lane position ($X$-axis of the graph) of different profiles was adjusted using the maximum of the monomer peak observed in all samples as a reference point. Ubiquitinated UNC-45 ($Ub_2$-$Ub_n$, Fig. 1) and ubiquitinated myosin (all ubiquitinated species, Fig. 7 and Supplementary Fig. 6) were quantified in ImageLab (BioRad). A box was drawn around the area of the lane showing the described ubiquitinated species and the HRP signal was integrated. To remove the background at the corresponding molecular weight (UNC-45-$Ub_2$-$Ub_n$ and myosin-$Ub_n$), the signal derived from lanes of control reactions without ATP, covering the same area, was subtracted from all measurements. Finally, the signal of the ubiquitinated species at different time points was plotted in Prism (GraphPad).

**Pull-down studies of UFD-2 with myosin and UNC-45.** All reactions were performed in assay buffer containing 20 mM Tris, pH 7.5, 50 mM NaCl at a final concentration of 5 μM protein. $His_6$-tagged UNC-45 proteins with or without myosin were pre-incubated at 27 °C for 60 min. Strep-UFD-2 was added, and the three proteins incubated at 30 °C for another 60 min to mimic the ubiquitination reaction conditions. Control reactions (shown in Supplementary Figs 3c and 5c) did not contain Strep-UFD-2. The reactions were applied to mini-spin columns containing 50 μl of equilibrated IBA Strep-Tactin Superflow beads. The beads were washed 5× with 500 μl assay buffer. Bound proteins were eluted in 6 × 25 μl steps in assay buffer supplemented with 2.5 mM d-desthiobiotin. Elution fractions 2–4 were pooled, applied to SDS-PAGE gels, and stained with Coomassie.

**C. elegans strains used in this study.** The C. elegans Bristol N2 strain was used as wild-type strain. The mutations used in this study are listed by chromosome as follows: LGI, chn-1(by155, deletion); LGII, ufd-2(tm1380, deletion); LGIII, unc-45(m94, mutation E781K), unc-45(b131, mutation G427E), unc-45(su2002, mutation L559S), and unc-45(e286, mutation L822F). All mutants were outcrossed at least four times to wild type prior to analysis/introduction of markers. For the generation of worms overexpressing UNC-45, the unc-45 cDNA containing a 3′ FLAG-tag sequence was subcloned into pPD30.38 (Andrew Fire lab, Stanford, CA) and 20 ng/μl of the resulting construct pPD30.38- $P_{unc-54}::unc-45^{FLAG}$ was co-injected with 20 ng/μl of the marker plasmid pPD114.108 (mec-7::GFP).

**C. elegans behavioral assay.** Worms were transferred without food to 14 cm NGM assay plates containing a cut out arena of Whatman filter paper soaked in 20 mM $CuCl_2$ to prevent them from leaving a 56 mm × 56 mm center area. A number of 60–100 adult animals (1 day post L4 larval stage), which had been removed from food 12 min prior to examination, were used in a single experiment. Each experiment was carried out three times.

Freely behaving animals were illuminated with flat red LED lights and recorded for 10 min at 3 fps on a 4 megapixel CCD camera (Jai) using Streampix software (Norpix). Movies were analyzed by customized MatLab-based image processing and tracking scripts based on software previously described[58]. The resulting trajectories were smoothed and used to calculate the instantaneous speed of the worm's centroid (1 s binning). Forward speed was obtained by excluding periods of backward locomotion (reversals) and turns. Deep omega turns and reversals were detected based on characteristic changes in object eccentricity and angular speed[59], and their frequencies were calculated in 15-s bins. Traces of behavioral time courses show mean and shaded SEM of all animals of all experiments. For quantification in Fig. 2d, the population mean for each individual experiment was calculated as the mean forward speed of all worms per assay over the whole recording, excluding the first and last 10 s.

**C. elegans crawling assay.** Wild-type, unc-45(m94), and unc-45(m94) ufd-2 (tm1380) worms were grown at 16 °C and transferred to 23 °C for 16, 22, or 30 h before the crawling assay. Individual young adult worms were placed on NGM agar plates seeded with E. coli OP50 and tracks were analyzed after 1 h as described in Hoppe et al[15]. A minimum of 10 worms were tested for each strain.

**Preparation of C. elegans samples for western blot analysis.** For preparing staged C. elegans extracts, the worms were synchronized at the L1 stage by treating mixed cultures with bleach and hatching embryos in the absence of food, and then grown in liquid culture at 20 °C until harvesting at 22 h (L3), 45 h (L4), and 70 h (adult). UNC-45 overexpressing worms were picked manually to ensure the presence of the extrachromosomal array. Worm lysates were subjected to western blot analysis using UNC-45, UNC-54 (mAb 28.2)[60], and PGK-1 (Invitrogen)-specific antibodies.

**Antibody generation.** Polyclonal antibodies against C. elegans $His_6$-UFD-2 and UNC-45-$His_6$ were generated by injecting recombinant protein into rabbits and collecting sera after several rounds of antigen boosting (performed by Gramsch laboratories). Antibodies were purified by applying sera onto an NHS-coupled

resin (GE Healthcare) charged with untagged UFD-2 or UNC-45-Strep. After extensive washing with PBS, the antibodies were eluted with 0.1 M Glycine pH 2.6, neutralized with 1 M Tris pH 8 and dialyzed into PBS containing 10% glycerol.

**Identification of UNC-45 ubiquitination sites by MS.** Ubiquitination reactions were performed as described above, applied to SDS-PAGE gels and then were stained with Coomassie. The four replicates shown in Supplementary Table 2 correspond to four reactions. Bands corresponding to mono-ubiquitinated UNC-45 were excised and after in-gel digest with trypsin, chymotrypsin, or subtilisin, the samples were extracted and used for MS analysis.

**Preparation of C. elegans lysates and immunoprecipitation.** Synchronous L1 worms were grown in liquid culture for 45 h at 20 °C. Upon reaching the L4 stage, the worms were grown for an additional 2.5 h at 34 °C and recovered for 2 h at 16 °C before harvesting. Worm pellets were washed and finally resuspended in 20 mM HEPES, pH 7.5, 150 mM KCl, 1 mM EDTA, 10% glycerol (IP buffer) supplemented with COMPLETE protease inhibitors (Roche). UNC-45 and UFD-2 antibodies as well as an unspecific anti-rabbit antibody were chemically cross-linked to ProteinA-Dynabeads (Life technologies) using DMP (dimethyl pimelimidate). Cleared C. elegans lysate, prepared by grinding frozen worms and subsequent centrifugation, was applied to the antibody-coupled beads. After incubation for 1 h at 4 °C the beads were washed with 20 mM HEPES, pH 7.5, 300 mM KCl, 1 mM EDTA, 10% glycerol, 0.05% NP-40, and IP buffer. Finally, antibody-bound protein complexes were eluted with 0.2 M glycine pH 2.0, eluates neutralized with 1 M Tris pH 9.0, and subjected to MS analysis.

**Sample preparation for NanoLC-MS analysis.** Elution fractions from IP experiments were reduced with DTT (dithiothreitol), alkylated with IAA (iodoacetamide), and enzymatically cleaved with trypsin (Promega Gold) o/n at 37 °C. The peptide solution was acidified with TFA to stop the enzymatic reaction and stored at −80 °C prior to the nanoLC-MS analysis.

**NanoLC-MS analysis and data analysis.** MS analysis of ubiquitination sites and the coIP protein sample were conducted on different systems, as described below.

For the IP experiment shown in Supplementary Table 1, tryptic peptides were separated by using an UltiMate 3000 RSLC Nano system (Thermo Fisher Scientific, Austria), equipped with a Nanospray Flex$^{TM}$ Ion Source (Thermo Fisher Scientific, Austria), and coupled to a Q Exactive mass spectrometer (Thermo Fisher Scientific, Austria). First, tryptic peptides were concentrated by a trap column (Thermo Fisher Scientific, PepMap C18, 5 mm × 300 μm ID) that was operated at a temperature of 30 °C and a flow rate of 25 μl $min^{-1}$, using 0.1% TFA as mobile phase. After 10 min, the trap column was switched in line with the analytical column (Thermo Fisher Scientific, PepMap C18, 500 mm × 75 μm ID, 3 μm). A 165 min gradient from buffer A (water/formic acid, 99.9/0.1 v/v) to B (water/acetonitrile/formic acid, 19.92/80/0.08 v/v/v) was applied to elute peptides. The LTQ Orbitrap Velos was operated in data-dependent mode, using a full scan in the Orbitrap (m/z range 380–1650, nominal resolution of 60,000, target value 1E6) followed by MS/MS scans of the 12 most abundant ions in the linear ion trap. MS/MS spectra (isolation width 2, target value 1E4, normalized collision energy 35%; activation value q 0.25) were acquired, and subsequent activation was performed through multistage activation. The neutral loss mass list was set to −98, −49, and −32.6 m/z, respectively. Dynamic exclusion was set to 60 s, and only charge states 2 and higher were accepted. For peptide identification, the RAW files were loaded into Proteome Discoverer (version 1.4.0.288, Thermo Scientific). All hereby created MS/MS spectra were searched using Mascot 2.2.07 (Matrix Science, London, UK) against the Uniprot protein sequence database (ver. 20150215; 90,860,905 sequences), using the taxonomy C. elegans (23,522 sequences). The following search parameters were used: beta-methylthiolation on cysteine was set as a fixed modification, oxidation on methionine, acetylation on lysine and protein-N-terminus, phosphorylation on serine, threonine, and tyrosine, and ubiquitination on lysine were set as variable modifications. Monoisotopic masses were searched within unrestricted protein masses for tryptic peptides. The peptide mass tolerance was set to ± 5 ppm and the fragment mass tolerance to ± 30 mmu. The maximal number of missed cleavages was set to 2. The result was filtered to 1% FDR using Percolator algorithm integrated in Proteome Discoverer. The localization of the phosphorylation sites within the peptides was performed with the tool phosphoRS[61].

For the mapping of Ub sites shown in Supplementary Table 2, the nano HPLC system used was an UltiMate 3000 Dual Gradient HPLC system (Dionex, Amsterdam, The Netherlands), equipped with a Proxeon nanospray source (Proxeon, Odense, Denmark), coupled to an LTQ-FT mass spectrometer (Thermo Fisher Scientific, Bremen, Germany). Peptides were loaded onto a trap column (Dionex PepMap C18, 5 mm × 300 μm ID, 5 μm particles, 100 Å pore size) at a flow rate of 25 μL $min^{-1}$ using 0.1% TFA as mobile phase. After 15 min, the trap column was switched in line with the analytical column (Dionex PepMap C18, 250 mm × 75 μm ID, 3 μm, 100 Å). Peptides were eluted using a flow rate of 275 nl $min^{-1}$, and a ternary gradient, as described[62] was used, with the following mobile phases: A (water/acetonitrile/formic acid, 95/5/0.1, v/v/v), B (water/acetonitrile/formic acid, 70/30/0.08, v/v/v), and C (water/acetonitrile/trifluoroethanol/formic acid, 10/80/

10/0.08, v/v/v/v). The total run time was 120 min. The LTQ FT was operated in data-dependent mode using a full scan in the ICR cell ($m/z$ range 400–1800, nominal resolution of 100,000 at $m/z$ 400, ICR target value 500,000) followed by MS/MS scans of the five most abundant ions in the linear ion trap. MS/MS spectra (normalized collision energy, 35%; activation value $q$ 0.25; activation time 30 ms; isolation width ± 3 Da) were acquired in the multistage activation mode, where subsequent activation was performed on fragment ions resulting from the neutral loss of −98, −49, or −32.6 $m/z$. Precursor ions selected for fragmentation (charge state 2 and higher) were put on a dynamic exclusion list for 180 s. Monoisotopic precursor selection was enabled. Additionally, singly charged parent ions were excluded from selection for MS/MS experiments and the monoisotopic precursor selection feature was enabled. For peptide identification, all MS/MS spectra were searched using Mascot 2.2.04 (Matrix Science, London, UK) against a custom-made protein sequence database containing the proteins of interest and contaminants. The generation of dta-files for Mascot was performed using the Extract MSn program (version 4.0, Thermo Scientific). The following search parameters were used: carbamidomethylation on cysteine was set as a fixed modification, oxidation on methionine, and ubiquitination on lysine were set as variable modifications. The peptide mass tolerance was set to ± 5 ppm and the fragment mass tolerance to ± 0.5 Da. The maximal number of missed cleavages was set to 2. Monoisotopic masses were searched within unrestricted protein masses for tryptic enzymatic specificity. Post-translationally modified peptides were filtered by a minimum Mascot Ions Score of 15 and validated by manual inspection.

**PRM of UNC-45 and UFD-2 proteins.** The nano HPLC system used was an UltiMate 3000 RSLC nano system (Thermo Fisher Scientific) coupled to a Q Exactive HF mass spectrometer (Thermo Fisher Scientific) equipped with a Proxeon nanospray source (Thermo Fisher Scientific). The nano HPLC system operated under same condition as in described above. The Q Exactive HF mass spectrometer was operating in scheduled PRM mode using low-resolution Full scan ($m/z$ range 370–1000, nominal resolution of 15,000, target value 3E6) and the maximum of 6 following PRM scans (nominal resolution 30,000, isolation window 1.6 $m/z$, maximum IT 650 ms and AGC target 1E5). Based on the shotgun analysis data, the unique peptides of both proteins of interest were manually selected. Supplementary Table 4 summarizes detailed information for selected peptides including $m/z$, peptide sequence, charge, scheduling window, and NCE. Acquired PRM data were processed in Skyline software (64-bit, v 3.7)[63] and manually verified. Final protein areas were calculated as the sum of individual peptide areas, which consist of areas of all unambiguously detected peptide fragments. Peptide fragments selected for the calculation co-eluted at the same retention time and they did not have any interfering signal in 5 min around the peak apex.

**Data availability.** Coordinates and structure factors of UNC-45$_{717}$ have been deposited in the Protein Data Bank under accession code 5mzu. Other data are available from the corresponding authors upon reasonable request.

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

## Acknowledgements

We thank the VBCF Protech Facility for the support with biophysical measurements; the Protein Chemistry Facility (Susanne Opravil, Ines Steinmacher, and Otto Hudez) for the MS analyses; Daniela Grzadziela, Ingrid Hums, and Manuel Zimmer for help with *C. elegans* behavioral assays; Luiza Deczsz, Robert Kurzbauer, and Nina Franicevic for support with initial biochemical experiments; Andrea Pichler for providing the hUba1 expression plasmid; the Epstein laboratory for the MHC-B specific antibody; Anton Meinhart for the support in the structural analysis; and the beamline staff at the ESRF for assistance during data collection. *C. elegans* strains were obtained from the Cae-norhabditis Genetics Center. This work was supported by the Austrian Research Pro-motion Agency (FFG) and FWF grants P22750 to D.H. and L.G., and Y597-B20 to A.D. The IMP is funded by Boehringer Ingelheim.

## Author contributions

D.H. and A.L. performed the protein biochemistry; D.H., L.G., and T.C. performed the structural analysis; M.R., D.H., and A.D. performed the *C. elegans* experiments; and K.S., R.I., and K.M. performed the MS analysis. T.C. outlined the work and wrote the manuscript with D.H.

## Additional information

**Competing interests:** The authors declare no competing financial interests.

