## [Peer Review File · Nature Communications]

Reviewers' comments:

Reviewer #1 (Remarks to the Author):

In the present manuscript, Hellerschmied et al. provide a detailed assessment of the role of the ubiquitin ligase UFD-2 and its interaction with the myosin chaperone UNC-45 and myosin itself.

As part of the comprehensive characterization of UFD-2 presented in this work, the authors first show that UFD-2 alone is able to polyubiquitinate its substrate, UNC-45, in absence of its putative co-ligase, CHN-1. Following this observation, the influence of CHN-1 and UFD-2 on motility of *C. elegans* was examined. Immunoprecipitation of both UNC-45 and UFD-2 followed by mass spectrometry (MS) analysis confirmed the interaction between the two proteins in heat-shocked worms. Site-specific (poly)ubiquitination of UNC-45 by UFD-2 was characterized by MS and revealed a characteristic modification pattern in two regions of the protein, near the myosin-binding region (the UCS domain) and the C-terminus. The specificity of the ubiquitination was further confirmed and the crystal structure of the UCS domain solved. Structural and biochemical data highlighted the role of unfolding of the UCS domain of UNC-45 by mutations that enhanced targeting by UFD-2. Finally, the authors showed that UFD-2 is able to polyubiquitinate myosin when in complex with UNC-45.

In summary, the manuscript provides a wealth of biochemical, biophysical and structural data that evaluate the role of UFD-2 and contradict earlier reports on the dependence of UFD-2 on CHN-1.

I have been asked to assess the sections of the manuscript that deal with MS data in particular. In this respect, I can confirm that the procedures are mostly in line with established standards in the field. However, essential information related to sample preparation procedures are absent in the manuscript. On p. 23-24, the method section covers IP experiments and MS analysis, respectively, but the processing steps from the elution of the proteins to the MS analysis are missing. This presumably includes proteolysis and purification steps that require more clarification.

Moreover, the overall design of the MS experiments could be explained in more detail. For example, Supplementary Table 2 lists four replicate experiments, but this information is not found anywhere else in the manuscript - information on the number of replicates should also be given for the characterization of the interactors (data from Suppl. Table 1).

Additional minor comments:

The version of database used should be given (p. 24).

Suppl. Table 1 mentions "Selected interacting proteins ... are listed". What were the criteria to include proteins in this table?

Reviewer #2 (Remarks to the Author):

Using an impressive combination of biochemistry, structural biology and *C. elegans* molecular genetics, Hellerschmied et al. report that they have uncovered a new mechanism of proteostasis in muscle cells. Specifically, they demonstrate that the myosin head chaperone UNC-45 can act as an adaptor to bring UFD-2, an E3 ligase, to polyubiquitinate unfolded myosin. In general, their data strongly supports the conclusion. These results and the model will be of interest to not only investigators studying muscle assembly/maintenance; they also will be of great interest to the entire myosin superfamily field, as the chaperone UNC-45 is involved in folding of both muscle and the many

types of non-muscle myosins. Publication is recommended if the authors can address the following issues:

1. Although the authors have discovered a new paradigm, the paper in general, seems to discount the possibility that UNC-45 not only identifies misfolded myosin and brings it to UFD-2 for its eventual degradation, but also acts as a chaperone to re-fold myosin heads. There is plenty of evidence that UNC-45 acts as a chaperone for refolding: (1) UNC-45 is required for embryonic muscle development (null alleles are Pat embryonic lethal; Venolia and Waterston, 1990); (2) reduced activity of UNC-45 (the ts mutant *unc-45(e286)* grown at the restrictive temperature) results in reduced numbers of thick filaments and decreased accumulation of MHC B (Barral et al. 1998); (3) their own data (Figure 2b) shows that UNC-45 protein levels are highest during larval stages, when the number of sarcomeres increases from 2 to 9 in each muscle cell; (4) UNC-45 can inhibit the thermal aggregation of myosin heads in vitro (Barral et al., 2002); and (5) a clever set of single molecule experiments suggesting that UNC-45 promotes the folding of the myosin head (Kaiser et al 2012). This should be discussed in the Introduction and the Discussion. They should also comment on what they think is the relative contribution of these two activities of UNC-45 (bringing myosin heads for ubiquitination vs. refolding myosin heads) in a muscle cell.
2. Although the nematode motility assays are sophisticated and well-done, in general, the use of *C. elegans* mutants is somewhat superficial. It is concerning that some of their phenotypic effects (motility, levels of UNC-45 etc.) might be due to background mutations as only single alleles of each gene were used, and although these may be null alleles, there was no mention as to whether the mutant alleles that were obtained were outcrossed to wild type (outcrossing 3-5X is standard in the field). "tm" alleles (like their *ufd-2(tm1380)*), when obtained from Japan, have not been outcrossed and usually have many background mutations.
3. For Figure 2a, for the UNC-45(OE) lanes, a loading control (e.g. PGK-1) should be shown. Could the authors comment on why in *ufd-2(tm1380)*, the level of UNC-45 seems to go down, rather than stay the same or increase?
4. The most puzzling result is that in the ts mutant *unc-45(m94)* the level of UNC-45 protein is increased compared with wild type. We would expect that if anything, the level of UNC-45 protein would decrease in a ts *unc-45* mutant, as most ts mutants that are missense mutations result in misfolding and instability of the protein. Was the western blot analysis conducted on m94 grown at the restrictive temperature? (this should be stated). Could m94 be dominant or semi-dominant? (this should be stated) This question of dominance is prompted because as the authors know, Landsverk et al. (2007), have shown that overexpression of UNC-45 leads to essentially the same phenotype as loss of function of *unc-45*. What is the molecular nature of the mutation in m94? (should be stated) Finally, because of the unexpected result of increased UNC-45 protein, the authors should repeat the analysis with one or two additional *unc-45* mutant alleles, including western blot and double mutant analysis.
5. The UNC-45 immunoprecipitation experiment is also puzzling (Suppl. Table 1)—it shows that UFD-2 is co-IPed (good), but that many other myofilament proteins are co-IPed including many thin filament proteins (actins, troponins, UNC-87). There is no evidence that UNC-45 acts as a chaperone for thin filament proteins, and since ATP was not included in their IP buffer, these thin filament proteins probably showed up because the myosin heads were still bound to thin filaments (sort of an artifact). This should be mentioned in the Results and/or Discussion.
6. The statement on page 8 (lines 200-201), "...UFD-2 acts as a specific E3 ligase targeting only the UNC-45 protein from the same organism." is not logical since the authors did not test human UFD2 on human UNC-45b. Perhaps the authors should say that "so far, then, nematode UFD-2 acts as a specific E3 ligase targeting only nematode UNC-45."
7. To make it easier for the reader, please maintain the same orientation of UNC-45 crystal structures and schematics in Figures 4 and 5. For example, re-orient the structures in Figure 5 by 180 degrees.
8. Please add size markers to the gel shown in Figure 7a.
9. Page 12, line 324, in addition to reference 24, please also cite reference 1.

10. Page 12, lines 327-330, in which the authors conclude that the reason the UNC -45:myosin complex elutes at an apparent mw of 670 kDa (instead of the expected mw of 200 kDa for a complex of 2 monomers) is that "the bound myosin should be present in a largely unfolded state". That could be true, but another interpretation is that there are multimers of UNC -45 (probably 3) present, each bound with a myosin head. If their gel exclusion chromatography utilized conditions in which they expected that such multimers would be disrupted, this should be stated.

Reviewer #3 (Remarks to the Author):

It has been reported previously that UFD-2 could function as an E4 ligase, elongating ubiquitin chains on substrates. The novel aspect of this work is that the authors find a conflicting result where UFD-2 directly ubiquitinates and forms polyubiquitin chains on substrates, including UNC -45 or unfolded myosin via UNC-45 which assists in binding the substrate. They use *C. elegans* as a model system to demonstrate in-vivo that UFD-2 does not have a role in UNC-45 down regulation but does have a role in development of motility. They show an interaction between UNC -45 and UFD-2 using co-immunoprecipitation and proteomics. Additionally they report two UFD-2 specific ubiquitination regions in the UCS domain of UNC-45 using mass spec analysis. K to R mutations in these regions show preference for ubiquitination of the C-terminal of UNC-45. The authors report a crystal structure of UNC-45, which contains high B-factors in the C-terminal region, as well as a slightly different conformation of this region compared to a previously reported structure, indicating flexibility of the region targeted for ubiquitination. The authors further explore the stability of UNC -45, in particular the UCS domain and the K to R mutants within this domain using limited proteolysis and CD with thermal unfolding. These data taken together with mass spec analysis indicate that the UCS domain is the most unstable part of the protein and therefore fits with the domain being targeted for degradation by UFD-2 ubiquitination. This led the authors to explore the ability of UNC -45 to work with UFD-2 as an adaptor to bind and target unfolded proteins for degradation, which they demonstrate for unfolded myosin.

Points:

1. Overall the experiments used to interrogate direct UNC -45 ubiquitination by UFD-2 are of wide variety and the data are of good quality.
2. Regarding the in-vivo data, the UFD-2 deletion mutant does not affect the level of overexpressed UNC-45 compared to WT. Here the authors argue that UFD-2 is not critical for developmental regulation of UNC-45. The authors then mention in the discussion that they looked at the level of myosin and found that the same UFD-2 deletion mutant also did not affect myosin levels. The data for this experiment are also not shown. They argue that this is due to a redundancy in the pathway for myosin degradation involving numerous different E3 ligases. The authors need to clarify to readers whether or not there are other E3 ligases that play a role in UNC -45 regulation other than UFD-2. Either way, they should be consistent or more clear with arguments for obtaining similar results for both tested substrates. This may strengthen their argument that unfolded myosin is in fact the substrate for UFD-2 while UNC-45 is not.
3. In-vitro activity assays should be quantitated, in particular, those in figure 7 where the affect of UNC-45 as an adaptor for UFD-2 directed ubiquitination of an unfolded protein is described, which is the main focus of the title. Other figures include 1 and 4.
4. What is missing is an in-vitro experiment with recombinant UNC -45 and UFD-2, which is required to prove the direct interaction between UNC -45 and UFD-2 that is reported in their model. For example,

a pull down with one of the proteins immobilized, SPR or ITC. This may prove that there are no other adaptors required for this interaction and a direct binding interface is required. The authors suggest that the core TPR, central and neck domain contains the binding site for UFD-2 in figure 4 b. To further narrow the binding of UFD-2 down to this domain, a deltaUCS mutant and a UCS domain alone mutant of UNC-45 would be interesting test cases for the in-vitro interaction experiment.

5. The ubiquitination reaction of unfolded myosin by UFD-2 with the adaptor UNC-45 was reconstituted in figure 7c. To control for UNC-45/UFD-2 complex targeting unfolded myosin, a parallel experiment should be carried out with folded myosin. If the UNC-45/UFD-2 complex targets unfolded myosin then folded myosin should not be ubiquitinated. In the absence of these data, we do not know. These data should strengthen the claim that the UNC-45/UFD-2 complex targets unfolded protein.

6. The model in figure 8 proposes both unstable UNC-45 or unfolded myosin as substrates. To test this new model for UFD-2 activity, it would be of interest to test an UCS canyon K to R mutant(s), which destabilizes UNC-45, in the presence of unfolded myosin in order to see a switch between unfolded myosin ubiquitination and unstable UNC-45 ubiquitination. My concern is that western blots in figure 7 only probe for myosin, not UNC-45 also. Is UNC-45 also ubiquitinated in these reactions with either folded or unfolded myosin? If so, the model needs to be updated to reflect the results.

Reviewer #4 (Remarks to the Author):

In their manuscript "UFD-2 is an adaptor-assisted E3 ligase targeting unfolded proteins" Hellerschmied et al. try to establish *C. elegans* UFD-2 as a ubiquitin E3 ligase targeting unfolded myosin with the help of the myosin-chaperone UNC-45. The manuscript starts out by disproving existing data that UFD-2 regulates UNC-45 protein levels. These data are solid but not spectacular.

Subsequently, the low resolution (3.8 Å) crystal structure of UNC-45 is presented which, as one would expect, resembles an earlier crystal structure of the same protein from the same lab, hence it is not clear why this structure was solved at all and is presented in the context of this manuscript. There are also a variety of issue as far as the crystal structure is concerned: The $\langle I/\sigma I \rangle$ clearly suggest that the data extend to higher resolution, why were these data not included in the refinement? Unbiased electron density maps should be shown to convince the reader that there are indeed significant conformational changes in the C-terminal part of the UCS? Why is Fig. 5B showing the B-factor distribution of the earlier and not the authors' new structure? With an R(free) of ~32% is this the best possible structure the authors can obtain?

Based on the structure the authors try to group lysine residues, which are modified by ubiquitin as demonstrated in MS experiments, into C-terminal sites as well as sites bordering the myosin-binding canyon with the distal sites supposedly inhibiting UNC-45 polyubiquitination and the canyon-sites stimulating polyubiquitination when the respective lysine(s) is(are) mutated to arginine. While a stimulation for the K704/706/713/717R is supported by the data, this is clearly not the case for the K637R variant (here there is rather a reduction) with the K713/717R being unchanged. As far as the C-terminal sites are concerned no clear difference to the wild-type can be detected. By the way, for the second gel in Fig. 4c the wild-type protein should be included on the same gel.

The authors then suggest that the K704/706/713/717R variant actually unfolds the UCS of UNC-45 (Fig. 6), however, their own CD data (Fig. S4) clearly indicate that unfolding of UNC-45 is always a two-state process for the wild-type and all mutants. If the UCS would be so easily destabilized, there should be two transitions at least for the wild-type protein with the UCS unfolding first and the rest of the protein melting at a higher temperature. By the way, how did the authors derive the experimental

errors for the individual measurements? I believe these experiments should be performed in triplicates (including biological replicates) followed by the calculations of the mean unfolding temperatures and their standard deviations. The errors presented in Fig. S4b are unrealistically low and suspiciously similar.

With respect to the final set of data (Fig. 7) the difference between UNC-45 WT and UNC-45 Delta UCS is augmented by the fact that there is simply more myosin in the +UNC-45 panels compared to myosin only and myosin with UNC-45 Delta UCS. This illustrates a general limitation of this manuscript where subtle changes in the ubiquitination pattern of a protein are used to draw general conclusions while no attempts are made to quantify these changes. Also, in the discussion the authors claim that UFD-2 ubiquitinates damaged myosin, however, nowhere in the manuscript are there any data that would support a specific recognition of damaged myosin; the data only demonstrate an activity vs. unfolded myosin which could represent newly synthesized but misfolded protein.

In summary, the manuscript suffers from insufficient quantification of the ubiquitination patterns of selected proteins, where in the eyes of this reviewer sometimes opposite effects are detected. Hence the main conclusions of this manuscript are not always supported by the experimental data.

Response to Reviewers

We would like to thank all four referees for their expert analysis and thoughtful comments. In preparing this revision, we have sought to incorporate their suggestions as far as possible. We believe our manuscript is substantially improved by these changes including the addition of new data to support our proposed model for the function of the UFD-2/UNC-45 complex. We hope to have thereby addressed any remaining concerns of the reviewers to their satisfaction. Below is our detailed point-by-point response, along with the reviewers' original comments in full.

Reviewer #1 (Remarks to the Author):

In the present manuscript, Hellerschmied et al. provide a detailed assessment of the role of the ubiquitin ligase UFD-2 and its interaction with the myosin chaperone UNC-45 and myosin itself.

As part of the comprehensive characterization of UFD-2 presented in this work, the authors first show that UFD-2 alone is able to polyubiquitinate its substrate, UNC-45, in absence of its putative co-ligase, CHN-1. Following this observation, the influence of CHN-1 and UFD-2 on motility of *C. elegans* was examined. Immunoprecipitation of both UNC-45 and UFD-2 followed by mass spectrometry (MS) analysis confirmed the interaction between the two proteins in heat-shocked worms. Site-specific (poly)ubiquitination of UNC-45 by UFD-2 was characterized by MS and revealed a characteristic modification pattern in two regions of the protein, near the myosin-binding region (the UCS domain) and the C-terminus. The specificity of the ubiquitination was further confirmed and the crystal structure of the UCS domain solved. Structural and biochemical data highlighted the role of unfolding of the UCS domain of UNC-45 by mutations that enhanced targeting by UFD-2. Finally, the authors showed that UFD-2 is able to polyubiquitinate myosin when in complex with UNC-45.

In summary, the manuscript provides a wealth of biochemical, biophysical and structural data that evaluate the role of UFD-2 and contradict earlier reports on the dependence of UFD-2 on CHN-1.

I have been asked to assess the sections of the manuscript that deal with MS data in particular. In this respect, I can confirm that the procedures are mostly in line with established standards in the field.

1) However, essential information related to sample preparation procedures are absent in the manuscript. On p. 23-24, the method section covers IP experiments and MS analysis, respectively, but the processing steps from the elution of the proteins to the MS analysis are missing. This presumably includes proteolysis and purification steps that require more clarification.

We now provide a detailed description of the MS procedure, including the sample preparation step. Sample preparation was summarized in the following way:

“Sample preparation for NanoLC-MS analysis: Elution fractions from IP experiments were reduced with DTT (dithiothreitol), alkylated with IAA (iodoacetamide) and enzymatically cleaved with trypsin (Promega Gold) o/n at 37°C. The peptide solution was acidified with TFA to stop the enzymatic reaction and stored at -80°C prior the nanoLC-MS analysis.” (page 26)

2) Moreover, the overall design of the MS experiments could be explained in more detail. For example, Supplementary Table 2 lists four replicate experiments, but this information is not found anywhere else in the manuscript - information on the number of replicates should also be given for the characterization of the interactors (data from Suppl. Table 1).

The replicates in **Supplementary Table 2** correspond to separate ubiquitination reactions, as now detailed in the Methods section. We further include additional details on how the MS experiment monitoring the UNC-45 ubiquitination sites was performed (page 25-27). Regarding the identification of UFD-2 and UNC-45 interaction partners presented in **Supplementary Table 1**, this was done on single UNC-45 and UFD-2 IP samples, as now indicated in the legend. To validate the putative UFD-2/UNC-45 interaction *in vivo*, we applied a more sophisticated, targeted MS approach, using a Q Exactive HF mass spectrometer to evaluate the protein-protein interaction in a semi-quantitative manner. To this end, selected peptides from the co-IP'ed UFD-2 and UNC-45 (**Supplementary Table 4**) were followed and quantified by Parallel-Reaction-Monitoring (PRM). The PRM analysis of samples from two replicate IP experiments clearly confirm the interaction between the two proteins (**Fig. 3d**). Moreover, the data show that the co-IPed proteins were present in markedly reduced amounts as compared to the UFD-2 and UNC-45 baits suggesting that the observed interaction is relatively weak, in agreement with the results of the *in vitro* pull-down experiments. In conclusion, the acquired MS data provide strong support for the interaction of UFD-2 and UNC-45 *in vivo*, as described in the following way:

„MS analysis of co-immunoprecipitated (coIP'ed) proteins revealed that UNC-45 and MHC-B (myosin heavy chain B, UNC-54) are interaction partners of UFD-2 *in vivo* (**Supplementary Table 1**). Moreover, when UNC-45 was immunoprecipitated from the same sample, UFD-2 as well as other muscle proteins could be detected in the elution fraction. Though we cannot exclude indirect binding, the interaction of UNC-45 and UFD-2, two non-sarcomeric proteins, could be functionally significant. To corroborate this interaction, we used the targeted MS approach of parallel-reaction monitoring (PRM)³³ as a more sensitive method to quantify UNC-45 and UFD-2 peptides in two additional IP samples (**Fig. 3d**). For this purpose, we selected peptides, identified in the original coIP experiment (**Supplementary Table 4**), for monitoring UNC-45 and UFD-2 abundance in the two samples. Our results strongly suggest that UFD-2 and UNC-45 directly interact in *C. elegans* muscle cells. Nevertheless, the small number of coIP'ed peptides compared to the bait protein indicate that the observed UFD-2/UNC-45 complex is relatively weak, pointing to a transient interaction of the E3 ligase and myosin chaperone.“ (page 7/8)

Additional minor comments:

a) The version of **database used should be given** (p. 24).

The version of the database is now listed in the methods section:

“All hereby created MS/MS spectra were searched using Mascot 2.2.07 (Matrix Science, London, UK) against the Uniprot protein sequence database (ver. 20150215; 90,860,905 sequences), using the taxonomy *Caenorhabditis elegans* (23,522 sequences).” (page 27)

b) Suppl. Table 1 mentions "Selected interacting proteins ... are listed". What were the criteria to include proteins in this table?

Selection criteria are now defined in the legend to **Supplementary Table 1**:

“The most abundant proteins based on peak area in each immunoprecipitate are listed in order of abundance in the UFD-2 IP sample.” (Supplementary information, page 10)

Reviewer #2 (Remarks to the Author):

Using an impressive combination of biochemistry, structural biology and *C. elegans* molecular genetics, Hellerschmied et al. report that they have uncovered a new mechanism of proteostasis in muscle cells. Specifically, they demonstrate that the myosin head chaperone UNC-45 can act as an adaptor to bring UFD-2, an E3 ligase, to polyubiquitinate unfolded myosin. In general, their data strongly supports the conclusion. These results and the model will be of interest to not only investigators studying muscle assembly/maintenance; they also will be of great interest to the entire myosin superfamily field, as the chaperone UNC-45 is involved in folding of both muscle and the many types of non-muscle myosins. Publication is recommended if the authors can address the following issues:

1. Although the authors have discovered a new paradigm, the paper in general, seems to discount the possibility that UNC-45 not only identifies misfolded myosin and brings it to UFD-2 for its eventual degradation, but also acts as a chaperone to re-fold myosin heads. There is plenty of evidence that UNC-45 acts as a chaperone for refolding: (1) UNC-45 is required for embryonic muscle development (null alleles are Pat embryonic lethal; Venolia and Waterston, 1990); (2) reduced activity of UNC-45 (the *ts* mutant *unc-45(e286)* grown at the restrictive temperature) results in reduced numbers of thick filaments and decreased accumulation of MHC B (Barral et al. 1998); (3) their own data (Figure 2b) shows that UNC-45 protein levels are highest during larval stages, when the number of sarcomeres increases from 2 to 9 in each muscle cell; (4) UNC-45 can inhibit the thermal aggregation of myosin heads in vitro (Barral et al., 2002); and (5) a clever set of single molecule experiments suggesting that UNC-45 promotes the folding of the myosin head (Kaiser et al 2012). This should be discussed in the Introduction and the Discussion. They should also comment on what they think is the relative contribution of these two activities of UNC-45 (bringing myosin heads for ubiquitination vs. refolding myosin heads) in a muscle cell.

We apologize if we have not described the chaperone function of UNC-45 in an appropriate way. In fact, we consider both pathways, myosin maturation and degradation, equally important, with UNC-45 acting as a central hub in this ‘triage’ decision. For a better-balanced description in the text, we extended the description of the chaperone role of UNC-45 in the Introduction, including references to the indicated studies. Moreover, we added a paragraph in the Discussion describing the dual activities of UNC-45 as a folding factor and as part of a potential myosin degradation pathway. With regards to the ‘relative contribution’ of these activities, a quantitative statement is currently not possible. However, we added a short paragraph, in which the dual activities are set in their biological context, i.e. that the chaperone activity is expected to be more relevant during development, whereas the quality control function may be more important in adult animals, in particular during stress situations when potentially damaged myosin molecules need to be removed. The revised paragraphs in the Introduction and Discussion read as follows:

“The importance of UNC-45 for muscle development was first demonstrated in the nematode *Caenorhabditis elegans*, where a number of UNC-45 *temperature-sensitive (ts)* mutants have

been identified^{1, 4, 5, 6}. When grown at the restrictive temperature, *ts* worms display a reduced number of thick filaments in their body wall muscles and exhibit an *uncoordinated* (*unc*) phenotype¹. Further studies in *Drosophila* and zebrafish established that UNC-45 (Unc45b in vertebrates) is important for skeletal and cardiac muscle development^{2, 3, 7}. Specifically, UNC-45 was shown to promote the folding of the motor domain of myosin molecules^{8, 9, 10}. In this process, UNC-45 acts as a co-chaperone together with the general chaperones Hsp70 and Hsp90^{8, 11, 12}.” (page 3)

“As the UNC-45 TPR domain is the major interaction site for both UFD-2 and Hsp70/Hsp90, it is likely that these interaction partners compete with each other, thereby determining the fate of the myosin bound to UNC-45. We propose that during development, the interplay with Hsp70 and Hsp90 is favored, promoting myosin folding and thick filament assembly. In contrast, during stress conditions UNC-45 may preferentially interact with UFD-2 targeting, in this case, misfolded myosin molecules for degradation.” (page 16)

2. Although the nematode motility assays are sophisticated and well-done, in general, the use of *C. elegans* mutants is somewhat superficial. It is concerning that some of their phenotypic effects (motility, levels of UNC-45 etc.) might be due to background mutations as only single alleles of each gene were used, and although these may be null alleles, there was no mention as to whether the mutant alleles that were obtained were outcrossed to wild type (outcrossing 3-5X is standard in the field). “tm” alleles (like their *ufd-2(tm1380)*), when obtained from Japan, have not been outcrossed and usually have many background mutations.

All mutants used in this study have been outcrossed at least 4 times, as now mentioned in the manuscript. We focused on these specific *C. elegans* alleles because they form the basis of most of the previously published work on UFD-2, CHN-1 and UNC-45, facilitating a comparison with those studies. The isolation and characterization of these alleles was originally reported in Hoppe et al (Cell, 2004), Janiesch et al (Nat Cell Biology, 2007) and Venolia et al (Cell Motil Cytoskeleton, 1999). The following information was added to the Methods section:

“The *C. elegans* Bristol N2 strain was used as wild-type strain. The mutations used in this study are listed by chromosome as follows: LGI, *chn-1(by155, deletion)*; LGII, *ufd-2(tm1380, deletion)*; LGIII, *unc-45(m94, mutation E781K)*, *unc-45(b131, mutation G427E)*, *unc-45(su2002, mutation L559S)*, and *unc-45(e286, mutation L822F)*. All mutants were outcrossed at least 4 times to wild-type prior to analysis/introduction of markers.” (page 23)

3. For Figure 2a, for the UNC-45(OE) lanes, a loading control (e.g. PGK-1) should be shown. Could the authors comment on why in *ufd-2(tm1380)*, the level of UNC-45 seems to go down, rather than stay the same or increase?

A loading control is now included in **Fig. 2a**. Regarding the apparent decrease in UNC-45 in *ufd-2* mutants noted by the reviewer, this minor change was not observed in a repeat of this experiment and seems to be a blotting artifact (see below). We have therefore replaced the panel accordingly (left: original panel; right: new panel):

4. The most puzzling result is that in the *ts* mutant *unc-45(m94)* the level of UNC-45 protein is increased compared with wild type. We would expect that if anything, the level of UNC-45 protein would decrease in a *ts* *unc-45* mutant, as most *ts* mutants that are missense mutations result in misfolding and instability of the protein. Was the western blot analysis conducted on m94 grown at the restrictive temperature? (this should be stated). Could m94 be dominant or semi-dominant? (this should be stated) This question of dominance is prompted because as the authors know, Landsverk et al. (2007), have shown that overexpression of UNC-45 leads to essentially the same phenotype as loss of function of *unc-45*. What is the molecular nature of the mutation in m94? (should be stated) Finally, because of the unexpected result of increased UNC-45 protein, the authors should repeat the analysis with one or two additional *unc-45* mutant alleles, including western blot and double mutant analysis.

Regarding the *unc-45* *ts*-mutations, we would like to note that these are missense mutations affecting single residues in the UCS and neck domain (b131-G427E, su2002-L559S, m94-E781K, and e268-L882F), as now stated in the manuscript. All four mutations do not reduce protein stability, but yield folded protein that can be characterized in functional terms, and can be even crystallized (our unpublished data).

In the revised manuscript, we included the E781K *ts*-mutant in the *in vitro* characterization of the UFD-2 ligase activity, showing a specific interaction between the two proteins. Together with the obtained *in vivo* data, these results demonstrate that *unc-45* *ts*-mutants yield, possibly in contrast to other *ts*-variants, folded proteins, even at the restrictive temperature. Though the underlying molecular details of the *ts*-phenotype are still unclear, the accumulation of the mutations in the UCS domain imply that the mutations may hinder the interaction with the myosin substrate. Consistent with this notion, we observed that the protein levels of all four UNC-45 *ts* mutants are upregulated in *C. elegans* pointing to a possible compensatory mechanism or feedback loop (see **Supplementary Fig. 1**). With

regards to the dominant/semi-dominant phenotype, all *unc-45* *ts*-mutants are reported to be recessive (Venolia et al., Cell Motil Cytoskeleton 1999), a result we have confirmed for the m94 allele used in our study. The revised paragraph in the Results section now reads as follows:

“When analyzing the motility of *unc-45(m94)* worms in crawling assays, we could confirm that the lack of UFD-2 partially rescues the temperature-dependent *unc* phenotype. Moreover, our detailed, time-resolved analysis revealed that the lack of UFD-2 delays the onset of the *unc* phenotype in *unc-45(m94)* worms, but cannot prevent it over time (**Fig. 3a**). When we quantified the UNC-45 levels in the analyzed strains at restrictive conditions (23°C), we observed an increase of UNC-45_{E781K} in the *unc-45(m94)* *ts*-mutant strain (**Fig. 3b**). A similar phenotype of exhibiting elevated levels of the myosin chaperone was also observed for the related *ts*-mutant strains *unc-45(b131, mutation G427E)*, *unc-45(su2002, mutation L559S)* and *unc-45(e286, mutation L822F)*, pointing to a common compensatory mechanism (**Supplementary Fig. 1c**). Importantly, however, we did not detect any further stabilization of the UNC-45_{E781K} *ts*-mutant protein in the strain lacking UFD-2 (**Fig. 3b**). These in vivo data indicate that the recovery of motility upon loss of UFD-2 is not due to a stabilization of the UNC-45 protein.” (page 7)

5. The UNC-45 immunoprecipitation experiment is also puzzling (Suppl. Table 1)—it shows that UFD-2 is co-IPed (good), but that many other myofilament proteins are co-IPed including many thin filament proteins (actins, troponins, UNC-87). There is no evidence that UNC-45 acts as a chaperone for thin filament proteins, and since ATP was not included in their IP buffer, these thin filament proteins probably showed up because the myosin heads were still bound to thin filaments (sort of an artifact). This should be mentioned in the Results and/or Discussion.

We agree with reviewer 2 in this point. Based on this experiment, the co-IPed proteins may not necessarily be direct interactors and could be indirectly pulled down as part of a larger ‘sarcomeric complex’. We have included a corresponding statement now in the Results part (please also see our comments to Referee-1, point 2, where we describe our further MS analysis):

„MS analysis of co-immunoprecipitated (coIP’ed) proteins revealed that UNC-45 and MHC-B (myosin heavy chain B, UNC-54) are interaction partners of UFD-2 in vivo (**Supplementary Table 1**). Moreover, when UNC-45 was immunoprecipitated from the same sample, UFD-2 as well as other muscle proteins could be detected in the elution fraction. Though we cannot exclude indirect binding, the interaction of UNC-45 and UFD-2, two non-sarcomeric proteins, could be functionally significant. To corroborate this interaction, we used the targeted MS approach of parallel-reaction monitoring (PRM)³³ as a more sensitive method to quantify UNC-45 and UFD-2 peptides in two additional IP samples (**Fig. 3d**). For this purpose, we selected peptides, identified in the original coIP experiment (**Supplementary Table 4**), for monitoring UNC-45 and UFD-2 abundance in the two samples. Our results strongly suggest that UFD-2 and UNC-45 directly interact in *C. elegans* muscle cells. Nevertheless, the small number of coIP’ed peptides compared to the bait protein

indicate that the observed UFD-2/UNC-45 complex is relatively weak, pointing to a transient interaction of the E3 ligase and myosin chaperone.“ (page 7/8).

6. The statement on page 8 (lines 200-201), “...UFD-2 acts as a specific E3 ligase targeting only the UNC-45 protein from the same organism.” is not logical since the authors did not test human UFD2 on human UNC-45b. Perhaps the authors should say that “so far, then, nematode UFD-2 acts as a specific E3 ligase targeting only nematode UNC-45.”

We agree and have rephrased the sentence accordingly:

“These data show that despite the high enzyme concentrations used in the in vitro assay, *C. elegans* UFD-2 acts as an E3 ligase targeting the nematode UNC-45 protein in a specific manner.” (page 8)

7. To make it easier for the reader, please maintain the same orientation of UNC-45 crystal structures and schematics in Figures 4 and 5. For example, re-orient the structures in Figure 5 by 180 degrees.

The structure shown in top of Fig. 5a is shown in the same orientation as in Fig. 4, as is now mentioned in the Figure legend. Moreover, the 180-degree rotation of the structural alignment below, which provides a better view on the rearrangement of the UCS domain, is indicated. Moreover, Fig. 5c is now presented in similar orientation as in Fig. 4.

8. Please add size markers to the gel shown in Figure 7a.

Size markers were added.

9. Page 12, line 324, in addition to reference 24, please also cite reference 1.

The reference was added.

10. Page 12, lines 327-330, in which the authors conclude that the reason the UNC-45:myosin complex elutes at an apparent mw of 670 kDa (instead of the expected mw of 200 kDa for a complex of 2 monomers) is that “the bound myosin should be present in a largely unfolded state”. That could be true, but another interpretation is that there are multimers of UNC-45 (probably 3) present, each bound with a myosin head. If their gel exclusion chromatography utilized conditions in which they expected that such multimers would be disrupted, this should be stated.

It should be noted that UNC-45 multimers are only transiently formed and cannot be detected by SEC under any condition. It requires a sophisticated cross-linking approach or extremely high protein concentrations as applied in crystallization trials to visualize UNC-45 chains in vitro. Therefore, it is unlikely that the high-molecular weight species is connected with a UNC-45 multimer, also keeping in mind that the UNC-45/myosin complex was present in

stoichiometric 1:1 ratio. To further validate our hypothesis, we performed SEC analyses comparing the elution behavior of folded and unfolded myosin. Importantly, unfolded myosin elutes, according to its non-compact shape and in contrast to folded myosin, at an ‘early’ elution volume corresponding to about 600 kDa, i.e. close to the apparent molecular weight of the UNC-45/myosin complex. These data suggest that the observed complex represents an unfolded myosin bound to its cognate chaperone UNC-45, as now described in the following way:

“The UNC-45:myosin complex with a predicted mass of 200 kDa eluted early, in stoichiometric ratio, from the SEC column corresponding to an apparent molecular weight of about 670 kDa. To address the composition of this large complex, we analyzed the two components individually. While both proteins attained their compact, functional state at 4°C, the heat-treated, presumably unfolded myosin eluted at a similar elution volume as the UNC-45:myosin complex (**Supplementary Fig. 5a**). Our SEC data thus suggest that myosin is captured in a largely unfolded form in the complex with UNC-45, thus reflecting the preference of the chaperone to bind to non-native protein segments as previously predicted from the shape of the myosin-binding canyon¹¹.” (page 13).

Reviewer #3 (Remarks to the Author):

It has been reported previously that UFD-2 could function as an E4 ligase, elongating ubiquitin chains on substrates. The novel aspect of this work is that the authors find a conflicting result where UFD-2 directly ubiquitinates and forms polyubiquitin chains on substrates, including UNC-45 or unfolded myosin via UNC-45 which assists in binding the substrate. They use *C. elegans* as a model system to demonstrate in-vivo that UFD-2 does not have a role in UNC-45 down regulation but does have a role in development of motility. They show an interaction between UNC-45 and UFD-2 using co-immunoprecipitation and proteomics. Additionally they report two UFD-2 specific ubiquitination regions in the UCS domain of UNC-45 using mass spec analysis. K to R mutations in these regions show preference for ubiquitination of the C-terminal of UNC-45. The authors report a crystal structure of UNC-45, which contains high B-factors in the C-terminal region, as well as a slightly different conformation of this region compared to a previously reported structure, indicating flexibility of the region targeted for ubiquitination. The authors further explore the stability of UNC-45, in particular the UCS domain and the K to R mutants within this domain using limited proteolysis and CD with thermal unfolding. These data taken together with mass spec analysis indicate that the UCS domain is the most unstable part of the protein and therefore fits with the domain being targeted for degradation by UFD-2 ubiquitination. This led the authors to explore the ability of UNC-45 to work with UFD-2 as an adaptor to bind and target unfolded proteins for degradation, which they demonstrate for unfolded myosin.

Points:

1. Overall the experiments used to interrogate direct UNC-45 ubiquitination by UFD-2 are of wide variety and the data are of good quality.

We thank the reviewer for this very positive and encouraging statement.

2. Regarding the in-vivo data, the UFD-2 deletion mutant does not affect the level of overexpressed UNC-45 compared to WT. Here the authors argue that UFD-2 is not critical for developmental regulation of UNC-45. The authors then mention in the discussion that they looked at the level of myosin and found that the same UFD-2 deletion mutant also did not affect myosin levels. The data for this experiment are also not shown. They argue that this is due to a redundancy in the pathway for myosin degradation involving numerous different E3 ligases. The authors need to clarify to readers whether or not there are other E3 ligases that play a role in UNC-45 regulation other than UFD-2. Either way, they should be consistent or more clear with arguments for obtaining similar results for both tested substrates. This may strengthen their argument that unfolded myosin is in fact the substrate for UFD-2 while UNC-45 is not.

To address this point, we have sought to clarify the different nature of the two potential UFD-2 substrates in the text. Most importantly, one putative substrate, UNC-45, is subject to regulatory proteolysis, requiring a pronounced specificity on the part of the E3 ligase, both in terms of time (at a specific developmental stage) and substrate. Therefore, deleting the

involved ubiquitin ligase should cause a clearly visible/defined phenotype, as the redundancy in regulatory-proteolysis pathways is typically low. Such phenotype could be demonstrably not observed for UNC-45 following deletion of UFD-2. In contrast to regulatory proteolysis, protein-quality-control mechanisms rely on a dense network of chaperone and protease systems that are often assembled in a highly redundant way providing various pathways for degrading the same substrate. This scenario seems to be valid in the case of misfolded myosin that is targeted by several E3 ligases (MuRF, Atrogin-1, Ozz E3 ligases). Accordingly, it is difficult to observe a stabilizing effect on the substrate by deleting a single ubiquitin ligase. To properly describe the proteolytic (regulatory) pathways targeting UNC-45, we have added the following paragraph to the Discussion:

“We demonstrate that UFD-2 poly-ubiquitinates UNC-45 on its own, thus exhibiting bona-fide E3 ligase activity in vitro. Modification of UNC-45 further does not depend on CHN-1, arguing against an E4 activity in this case. Moreover, global protein levels of UNC-45 were not affected upon deleting the ubiquitin ligases in *C. elegans*. As the regulatory proteolysis of a certain substrate protein is known to rely on E3 ligases that exhibit pronounced substrate specificity and operate in a precisely defined spatio-temporal manner as for example demonstrated for Aurora kinases³⁸, our data strongly suggest that UFD-2 and CHN-1 are not involved in the developmental regulation of UNC-45. In fact, the E3 ligase controlling the levels of UNC-45 remains to be identified to fully understand the biological role and regulation of this myosin-specific chaperone.” (page 15)

With regards to the missing myosin experiment, we would like to note that this experiment was shown in **Fig. 3**. Possibly, the similar names UNC-45 and UNC-54 in the Figure annotation may have been misleading.

3. In-vitro activity assays should be quantitated, in particular, those in figure 7 where the affect of UNC-45 as an adaptor for UFD-2 directed ubiquitination of an unfolded protein is described, which is the main focus of the title. Other figures include 1 and 4.

To address this point (also raised by the reviewer 4, comment 4), we repeated all comparative ubiquitination assays to better quantify the labeled Ub marks in Western blots (**Fig. 1**, **Fig. 4**, **Fig. 7**). For quantification of the different ubiquitination reactions we plotted lane profiles using the program Fiji and overlaid them in Graphpad Prism for direct comparison. The amounts for the various Ub conjugates (Ub1, Ub2, Ub3, ..., poly-Ub) are clearly visible in the resulting plots. As can be seen in this quantification, CHN-1 and UFD-2 have entirely distinct ubiquitin ligase activities against the UNC-45 substrate (i.e. UFD-2 is a bona-fide E3, **Fig. 1**). Moreover, these plots highlight the specific targeting of the UCS domain by UFD-2 (**Fig. 4**) as well as the function of UNC-45 as a specific adaptor protein promoting UFD-2-mediated poly-ubiquitination of myosin (**Fig. 7**).

4. What is missing is an in-vitro experiment with recombinant UNC-45 and UFD-2, which is required to prove the direct interaction between UNC-45 and UFD-2 that is reported in their model. For example, a pull down with one of the proteins immobilized, SPR or ITC. This

may prove that there are no other adaptors required for this interaction and a direct binding interface is required. The authors suggest that the core TPR, central and neck domain contains the binding site for UFD-2 in figure 4 b. To further narrow the binding of UFD-2 down to this domain, a deltaUCS mutant and a UCS domain alone mutant of UNC-45 would be interesting test cases for the in-vitro interaction experiment.

We thank the reviewer for this excellent suggestion. In fact, we performed pull-down experiments using full-length UNC-45 and several deletion constructs. In addition to demonstrating the direct and specific interaction of UNC-45 and UFD-2, these experiments additionally reveal that UFD-2 primarily interacts with the TPR domain of UNC-45 (**Fig. 4c**). Thus, thanks to the reviewer's suggestion, we could improve and further refine our model of how UNC-45 and UFD-2 establish a composite ubiquitin ligase targeting myosin. The new data are described as follows:

“To further characterize the interaction between UNC-45 and UFD-2, we performed a pull-down analysis using differently tagged UFD-2 and UNC-45 proteins (**Fig. 4c, Supplementary Fig. 2c**). These in vitro experiments demonstrated that full-length UNC-45 can directly interact with the UFD-2 ubiquitin ligase. Moreover, we noted that the UNC-45_{GST-UCS} and UNC-45_{ΔTPR} deletion constructs no longer interacted with UFD-2, while deletion of the UCS domain resulted in a slightly weakened binding. Based on these data, the main binding site of UFD-2 seems to reside in the TPR domain of UNC-45, with the UCS domain stabilizing complex formation. In addition, the pull-down studies revealed that UFD-2 still interacts with the UNC-45_{E781K} *ts*-mutant, yet we observed a reduced ubiquitination of this mutant protein, possibly pointing to changes in the targeted UCS domain (**Fig. 4a**). Taken together, the results of the interaction studies and ubiquitination assays indicate that the TPR domain of UNC-45 serves as the major docking site licensing UFD-2 to access and poly-ubiquitinate the adjacent UCS domain.” (page 9).

“Strikingly, however, in the presence of UNC-45, myosin was poly-ubiquitinated by UFD-2 (**Fig. 7cd**). Consistent with these results, a direct interaction between UNC-45, myosin and UFD-2 can be observed in pull-down studies (**Fig. 7e, Supplementary Fig. S5d**).“ (page 14)

5. The ubiquitination reaction of unfolded myosin by UFD-2 with the adaptor UNC-45 was reconstituted in figure 7c. To control for UNC-45/UFD-2 complex targeting unfolded myosin, a parallel experiment should be carried out with folded myosin. If the UNC-45/UFD-2 complex targets unfolded myosin then folded myosin should not be ubiquitinated. In the absence of these data, we do not know. These data should strengthen the claim that the UNC-45/UFD-2 complex targets unfolded protein.

This important control experiment was already presented in the original manuscript, albeit buried in the Supplemental Data (**Fig. S5**). We now specifically refer to this Figure, emphasizing the finding that folded myosin cannot be directed by UNC-45 for UFD-2-mediated ubiquitination:

“Notably, the observed polyubiquitination was dependent on incubating UNC-45 and myosin at higher temperature (27°C), as mixing at 4°C prior to the ubiquitination reaction did not

yield any polyubiquitinated myosin molecules (**Supplementary Fig. 5c**). These data provide compelling evidence that UNC-45 recruits UFD-2 to poly-ubiquitinate unfolded myosin.” (page 14).

6. The model in figure 8 proposes both unstable UNC-45 or unfolded myosin as substrates. To test this new model for UFD-2 activity, it would be of interest to test an UCS canyon K to R mutant(s), which destabilizes UNC-45, in the presence of unfolded myosin in order to see a switch between unfolded myosin ubiquitination and unstable UNC-45 ubiquitination. My concern is that western blots in figure 7 only probe for myosin, not UNC-45 also. Is UNC-45 also ubiquitinated in these reactions with either folded or unfolded myosin? If so, the model needs to be updated to reflect the results.

Regarding the concern that the Western blot analysis only probed for myosin, we would like to note that we have also examined UNC-45 ubiquitination in the same assay (**Fig. 7c**). These data indicate that UFD-2 targets UNC-45 and myosin in parallel and that there is no switch in substrate preference when using folded or unfolded myosin. We have endeavored to make this point more clearly in the adapted model figure (**Fig. 8**):

“UNC-45 is a myosin chaperone that teams up with Hsp70/Hsp90 to promote the folding and assembly of muscle myosin (left). Our data shows that the UFD-2 ubiquitin ligase is another partner protein of UNC-45, re-directing the chaperone towards a ubiquitination pathway. Upon binding to its TPR domain (red), the UFD-2 E3 ligase gets properly positioned to poly-ubiquitinate a presented protein, either the UCS domain (grey) of UNC-45 itself or a UCS-bound myosin (green). Importantly, UFD-2 has a clear preference for marking unfolded proteins, as indicated. ” (corresponding Figure legend, page 39)

Reviewer #4 (Remarks to the Author):

In their manuscript "UFD-2 is an adaptor-assisted E3 ligase targeting unfolded proteins" Hellerschmied et al. try to establish *C. elegans* UFD-2 as a ubiquitin E3 ligase targeting unfolded myosin with the help of the myosin-chaperone UNC-45. The manuscript starts out by disproving existing data that UFD-2 regulates UNC-45 protein levels. These data are solid but not spectacular.

1) Subsequently, the low resolution (3.8 Å) crystal structure of UNC-45 is presented which, as one would expect, resembles an earlier crystal structure of the same protein from the same lab, hence it is not clear why this structure was solved at all and is presented in the context of this manuscript. There are also a variety of issue as far as the crystal structure is concerned: The clearly suggest that the data extend to higher resolution, why were these data not included in the refinement? Unbiased electron density maps should be shown to convince the reader that there are indeed significant conformational changes in the C-terminal part of the UCS? Why is Fig. 5B showing the B-factor distribution of the earlier and not the authors' new structure? With an R(free) of ~32% is this the best possible structure the authors can obtain?

Regarding the motivation of performing a structural analysis of the UCS domain of UNC-45, we now provide a more appropriate reasoning in the corresponding paragraph:

“To further characterize the structural motif targeted by the UFD-2 ligase, we performed a crystallographic analysis of the UCS domain of UNC-45. While extensive crystallization trials of the UCS domain containing various KR mutations were not successful, we succeeded in crystallizing the wild-type UNC-45 protein in a crystal form, capturing the UCS domain in a distinct state.” (page 10).

As for the ‘why’ of solving this particular crystal structure, it was our scientific curiosity to explain the peculiar biochemical properties of the UCS domain as the preferred substrate for UFD-2 that motivated us to visualize these properties by a structural analysis. Although the general domain organization is indeed similar to our original structural model (Gazda et al, 2013), the new data reveal that the UCS domain is capable of adopting significantly different but discrete conformations. The observed rearrangement of the newly identified sub-domains in the UCS fold account for about 10 Å and are thus far beyond the coordinate errors and highly significant. A prediction of such structural rearrangements and the presence of two UCS sub-domains was not previously possible. These new data will be invaluable in further characterizing the myosin-binding function of UNC-45 and addressing the biological roles of UCS proteins implicated in various myosin-dependent processes.

We understand the critical comments of the reviewer on the quality and resolution of our structure. From a technical point of view, the truncation of the diffraction data to 3.8 Å was intentional, although, as the reviewer noted, the crystals diffracted to higher resolution (indicated by the relatively high $\langle I/\sigma(I) \rangle$ of 3.8 in the last resolution shell). Owing to the unusually extended shape of the unit cell, collecting high-resolution diffraction data was, however, not possible, as the long 717 Å c-axis and the crystal morphology hindered orientating this axis ‘perpendicular’ to the X-ray beam. Therefore, completeness of the data

sharply dropped beyond the indicated resolution. Another point of concern by the referee relates to the refinement of the model at 3.8 Å, whether refinement has sufficiently converged with an R_{free} of 32%. It is well established that the R_{free} (similar to the R_{work}) is closely correlated with the phase error of the model (see Bruenger, Nature (1992), Kleywegt and Bruenger, Structure (1996)) and is thus also inherently connected with the resolution of a particular structure. Therefore, the presented UNC-45₇₁₇ structure should be compared with structures determined at similar resolution. Applying the criteria from Urzhumtsera et al. Acta Cryst (2009), there are no concerns regarding the refinement process: While the R_{free} is slightly higher than the median (29% for 655 structures at similar resolution), the value of 32% is absolutely within the acceptable range and, importantly, comes with the benefit of an excellent protein geometry, as stereochemical parameters are substantially better than those of the corresponding median values (rmsd for bonds of 0.006 Å in contrast to 0.009 Å, and a rmsd for angles of 0.9° in contrast to 1.3°). Given the comparatively low resolution of our crystal structure, we decided to pursue this conservative approach rather than overfitting our data. In conclusion, we are convinced that our model is sufficiently robust to characterize the gross conformational changes within the UCS domain described in the manuscript. To illustrate the quality of our model, we followed the advice of the referee and included a figure showing the electron density map of the entire UCS domain (**Supplementary Fig. 3**). We decided to use a 2mFo-DFc instead of a simulated-annealing composite omit map, although the latter map should be to some extent less biased. The reason for this is that following the simulated annealing protocol at 3.8 Å resolution severely worsened the refined geometry and thus also the phase information. Moreover, we decided to show the B-factor plot of UNC-45₈₁₅ in the main part, as this structure was determined at higher resolution and because the distribution of thermal motion factors is very similar with UNC-45₇₁₇, as now presented in the **Supplementary Fig. 3d**.

2) Based on the structure the authors try to group lysine residues, which are modified by ubiquitin as demonstrated in MS experiments, into C-terminal sites as well as sites bordering the myosin-binding canyon with the distal sites supposedly inhibiting UNC-45 polyubiquitination and the canyon-sites stimulating polyubiquitination when the respective lysine(s) is(are) mutated to arginine. While a stimulation for the K704/706/713/717R is supported by the data, this is clearly not the case for the K637R variant (here there is rather a reduction) with the K713/717R being unchanged. As far as the C-terminal sites are concerned no clear difference to the wild-type can be detected. By the way, for the second gel in Fig. 4c the wild-type protein should be included on the same gel.

As supposed by the reviewer, quantifying the individual ubiquitination patterns shows that the reduction of ubiquitination following mutation of the C-terminal sites is - although present - not as pronounced as implicated in the text. However, we would like to note that the differences in the observed poly-Ub pattern justifies grouping the different KR mutations into different classes. UNC-45 mutants in which residues bordering the myosin-binding canyon are exchanged are most efficiently targeted by UFD-2. Our data indicates that a destabilized and possibly unfolded UCS domain is the best substrate for UFD-2. To confirm this

hypothesis, we performed the following additional experiment: Upon conducting ubiquitination assays with pre-warmed UNC-45 (incubation for 60 min at 30°C, i.e. close to T_m to destabilize the UCS domain prior to ubiquitination), we observed an increase in poly-Ub conjugates for K704/706/713/717R and K713/717R mutants compared to wild type (**Fig. 4e**). Please note that all mutants and the wild-type protein are shown and compared on the same blot. We omitted the K637R mutant from this analysis, as this residue is not directly bordering the myosin-binding canyon, but is rather close to the extended UCS loop, a highly flexible structural motif also implicated in myosin binding. The revised paragraph now reads as follows:

“To further explore the substrate selectivity of UFD-2, we systematically exchanged the targeted lysine residues of UNC-45 with arginines (KR mutants) and analyzed the resultant mutants in ubiquitination assays. Unexpectedly, mutating lysine residues lining the myosin-binding canyon yielded a substrate that was better poly-ubiquitinated than wild-type UNC-45 (**Fig. 4e and Supplementary Fig. 2fg**). Quantitative analysis of the ubiquitination profiles clearly revealed an increase in the high-molecular weight poly-Ub UNC-45 for the K704/706/713/717R (KR_{canyon}) mutant, while the K637R and K938/943R mutants were not as strongly poly-ubiquitinated as wild-type UNC-45 (**Fig. 4e and Supplementary Fig. 2f**). These findings suggest that UFD-2 targets the UCS domain of UNC-45 in a topologically specific manner.” (page 10)

3) The authors then suggest that the K704/706/713/717R variant actually unfolds the UCS of UNC-45 (Fig. 6), however, their own CD data (Fig. S4) clearly indicate that unfolding of UNC-45 is always a two-state process for the wild-type and all mutants. If the UCS would be so easily destabilized, there should be two transitions at least for the wild-type protein with the UCS unfolding first and the rest of the protein melting at a higher temperature. By the way, how did the authors derive the experimental errors for the individual measurements? I believe these experiments should be performed in triplicates (including biological replicates) followed by the calculations of the mean unfolding temperatures and their standard deviations. The errors presented in Fig. S4b are unrealistically low and suspiciously similar.

When stating that UNC-45 unfolding is a two-state process, the reviewer may have been misled by the melting and aggregation events observed by CD (**Fig. S4**). However, multi-domain proteins typically unfold in a cooperative manner rather than the separate domains unfold independently. Accordingly, all studied UNC-45 variants exhibit a single T_m , which is heavily influenced by mutations in the UCS domain. Regarding the error reported for the CD data, the values shown in the original **Fig. S4b** were the errors of the curve fit. To avoid any confusion, we have removed these numbers. With regards to replicates, we have repeated these measurements twice, as is common practice in the field. To clarify our argumentation, we have revised the relevant paragraph in the following way:

“To corroborate the limited proteolysis data, we performed circular dichroism (CD) measurements. The recorded melting curve of wild-type UNC-45 revealed an unfolding step at about 35°C and a further transition occurring at 60-70°C due to aggregation of the protein (**Supplementary Fig. 4bcd**). Strikingly, an analysis of the individual T_m values (melting

temperature of unfolding step) revealed a clear correlation between protein stability and the efficiency in being ubiquitinated by UFD-2. The most efficiently targeted UNC-45 KR_{canyon} mutant proteins were also the most destabilized variants as indicated by a markedly lowered T_m value (33°C compared to 35°C of wild-type). In addition, we noted that deletion of the UCS domain generated a truncated protein having a higher T_m value (38°C) than the wild-type protein. These data indicate that the UCS domain represents the most unstable portion of the UNC-45 molecule.” (page 12)

Finally, we would like to point out that the unfolding process of UNC-45 was characterized by two different techniques, CD spectroscopy paired with limited proteolysis. Importantly, sequencing of the segment that was most susceptible to proteolytic cleavage revealed that the C-terminal half of the UCS domain (residues 621-961) is the most unstable portion of the UNC-45 molecule, confirming the data from CD spectroscopy and protein crystallography. Moreover, the limited proteolysis experiments showed that the C-terminal UCS fragment of the K704/706/713/717R and the K713/717R mutants is most rapidly digested, supporting the proposed grouping of UCS mutants.

4) With respect to the final set of data (Fig. 7) the difference between UNC-45 WT and UBC-45 Delta UCS is augmented by the fact that there is simply more myosin in the +UNC-45 panels compared to myosin only and myosin with UNC-45 Delta UCS. This illustrates a general limitation of this manuscript where subtle changes in the ubiquitination pattern of a protein are used to draw general conclusions while no attempts are made to quantify these changes. Also, in the discussion the authors claim that UFD-2 ubiquitinates damaged myosin, however, nowhere in the manuscript are there any data that would support a specific recognition of damaged myosin; the data only demonstrate an activity vs. unfolded myosin which could represent newly synthesized but misfolded protein.

Following the recommendation of this reviewer and reviewer 3, we have repeated all comparative ubiquitination analyses and quantified the individual ubiquitination patterns with regards to -Ub1, -Ub2, -Ub3 and poly-Ub events. Please see our comments to reviewer 3, point 3 for more details. We agree with reviewer 4 that we haven't used “damaged” myosin in our in vitro assays. The full substrate spectrum of UNC-45 and/or the UNC-45/UFD-2 ubiquitin ligase complex is not yet defined and will be an important subject for further research. We therefore omit the term “damaged” myosin when discussing UNC-45 substrates and instead refer to misfolded/unfolded myosin throughout the manuscript.

5) In summary, the manuscript suffers from insufficient quantification of the ubiquitination patterns of selected proteins, where in the eyes of this reviewer sometimes opposite effects are detected. Hence the main conclusions of this manuscript are not always supported by the experimental data.

As mentioned in our response to reviewer 3 (point 3) and reviewer 4 (point 4), we have now quantified the ubiquitination reactions in **Fig. 1**, **Fig. 4** and **Fig. 7c**. The corresponding plots illustrate the ability of UFD-2 to function as bona-fide E3 ligase, allowing us to compare the

activity of UFD-2 against different UNC-45 substrates and document that myosin poly-ubiquitination depends on the presence of a UFD-2/UNC-45 complex. Together, these quantitative data provide strong and unbiased evidence for our proposed model. We further include a detailed protein-protein interaction analysis, demonstrating that UFD-2 directly binds to UNC-45, using the TPR domain as main docking site. Importantly, this contact is also essential for myosin ubiquitination as shown in **Fig. 7c**. Finally, a further pull-down experiment visualizes the ternary complex formed by UFD-2, UNC-45 and misfolded myosin (**Fig. 7e**). Collectively, these data support our main conclusion that UFD-2 employs UNC-45 as an adaptor to poly-ubiquitinate unfolded myosin, uncovering a novel role for the myosin chaperone to maintain proteostasis in muscle cells.

Reviewers' comments:

Reviewer #2 (Remarks to the Author):

The authors have done a good job at addressing my previous concerns of their first submission. The overall quality has been improved also based on their responses to the other critiques. This fine paper is now ready for publication.

Reviewer #3 (Remarks to the Author):

The authors have addressed initial concerns regarding clarification of specific points, demonstration of direct interaction and the model. However the attempt to quantify the reactions is unsatisfactory. The reactions need to be quantified in comparison to standards within the linear range of the detection method and a reaction rate calculated for the ubiquitination reaction. The plots supplied do not report this information and therefore do not demonstrate reliability. In many cases, for example Figure 1 a and b, the plots are less useful than the western blots as only one time point is plotted. In addition, there are two ubiquitination reactions occurring in Figure 7 c and d, which need to be quantified. These experiments are required to supply reliable data as evidence for the claim of the manuscript that UFD-2 is an adaptor assisted E3 ligase mediating ubiquitination of unfolded proteins. The paper is not suitable for publication without this information.

We would like to thank the reviewers for their insightful comments. Please find below our response addressing the remaining point raised by Reviewer #3.

Reviewer #3 (Remarks to the Author): The authors have addressed initial concerns regarding clarification of specific points, demonstration of direct interaction and the model. However, the attempt to quantify the reactions is unsatisfactory. The reactions need to be quantified in comparison to standards within the linear range of the detection method and a reaction rate calculated for the ubiquitination reaction. The plots supplied do not report this information and therefore do not demonstrate reliability. In many cases, for example Figure 1 a and b, the plots are less useful than the western blots as only one time point is plotted. In addition, there are two ubiquitination reactions occurring in Figure 7 c and d, which need to be quantified. These experiments are required to supply reliable data as evidence for the claim of the manuscript that UFD-2 is an adaptor assisted E3 ligase mediating ubiquitination of unfolded proteins. The paper is not suitable for publication without this information.

We apologize if we did not adequately address this point.

With regards to the technical aspect of the quantification, we would like to note that protein bands in our blots were visualized using the BioRad ChemiDoc MP Imaging system. This system allows for quantitative analysis of chemiluminescent signal within the linear dynamic range of the instrument. Even in the absence of external standards, quantitative relative comparisons of ubiquitination reactions are therefore certainly possible. Further experimental details on our analysis are provided in the revised Methods section (p. 21/22).

In response to the reviewer's original criticism, we had repeated all ubiquitination experiments and measured corresponding Ub lane profiles. This analysis allowed a quantitative comparison of the different ubiquitinated protein species (-Ub1, -Ub2, -Ub3, -Ubn) in each reaction. We now include densitometric quantifications of the accumulation of high-molecular weight ubiquitin adducts at all time points. In the revised Fig. 1, this quantitative analysis clearly supports our conclusion that UFD-2 has bona-fide E3 ligase activity against UNC-45. In the revised Fig. 7, we quantified the ubiquitinated myosin fractions for all reactions. Plotting these data as time-courses allowed us to directly compare the activity of different UNC-45 forms as UFD-2 adaptors (all 6 reactions are now included). These data clearly show that the poly-ubiquitination of myosin by UFD-2 is dependent on UNC-45. In contrast, CHN-1 does not exhibit a stimulatory effect on UFD-2 when targeting myosin, as illustrated in the time-course analysis presented in Supplemental Fig. 5.

While we consider a detailed kinetic analysis of UFD-2 ligase activity (e.g. determining rate constants towards the different substrates) to be beyond the scope of the current study, our quantifications fully support the key assertions made: UFD-2 is a bonafide E3 ligase that is assisted by UNC-45 to target myosin.

Reviewers' comments:

Reviewer #3 (Remarks to the Author):

The Bio-Rad ChemiDoc assumes the samples are within the dynamic range of the instrument as long as the signal is not saturated. This does not determine whether the response is within a linear range in terms of the Western Blot signal and transfer. Therefore a standard curve or dilution series reports whether the response is reliable for quantitation.

Reviewer #3 (Remarks to the Author): The Bio-Rad ChemiDoc assumes the samples are within the dynamic range of the instrument as long as the signal is not saturated. This does not determine whether the response is within a linear range in terms of the Western Blot signal and transfer. Therefore a standard curve or dilution series reports whether the response is reliable for quantitation.

To address this point we have now included a dilution series of the respective ubiquitination reactions, which allowed us to generate standard curves (Supplementary Fig. 1, 3h and 6f). The additional panels are also depicted below.

Supplementary Figure 1

Supplementary Figure 3h

Supplementary Figure 6f

These data clearly demonstrate that the quantified HRP-signal is within the linear range of detection. The R^2 value of 0.99 for each of the standard curves indicates a good fit with the applied linear regression model and clearly shows the linear increase in HRP-signal with increasing amounts of standard ubiquitination reaction applied to the gel.

Plotting the HRP-signal obtained for the reactions onto the same graph shows that their signals are in the linear range of detection (with the K703, 706, 713, 717R mutant in Supplementary Fig. 3h close to the limit of the linear range).

A description of how these Western blots and standard curves were generated is included in the Materials and Methods section: *Quantification of in vitro ubiquitination products* (p. 21/22) as follows:

“Quantification of in vitro ubiquitination reactions was performed on Western blots developed using the ChemiDoc MP Imaging system (BioRad), which is designed for quantitative analysis of chemiluminescent signal. To additionally ensure that the quantified Western blot signal is within the linear range of detection, we generated standard curves using a dilution series of the respective ubiquitination reaction, which was incubated for 180 min at 30°C. The signal was quantified in ImageLab (Biorad) by drawing a box of the same size at the center of each lane showing ubiquitinated protein. The same size box was used to measure the background signal that was subtracted. The standard curve was generated in Prism (Graphpad) using the linear regression curve fitting model. The ubiquitination reactions in Supplementary Fig. 1, 3 and 6 were quantified in the same manner. The thereby acquired HRP-signals are within the linear range of the Western blot as demonstrated by plotting the obtained quantified signals onto the graph of the standard curve. ...”

These additional data corroborate the reliability of the quantifications of the in vitro ubiquitination assays and further support our conclusion that UFD-2 acts as an E3 ligase towards UNC-45 and an UNC-45/myosin complex.